# A Novel Hybrid High-Dimensional PSO Clustering Algorithm Based on the Cloud Model and Entropy

**Ren-Long Zhang and Xiao-Hong Liu ***

School of Management, Guizhou University, Guiyang 550025, China
* Correspondence: liuxiaohongsci@163.com

**Abstract:** With the increase in the number of high-dimensional data, the characteristic phenomenon of unbalanced distribution is increasingly presented in various big data applications. At the same time, most of the existing clustering and feature selection algorithms are based on maximizing the clustering accuracy. In addition, the hybrid approach can effectively solve the clustering problem of unbalanced data. Aiming at the shortcomings of the unbalanced data clustering algorithm, a hybrid high-dimensional multi-objective PSO clustering algorithm is proposed based on the cloud model and entropy (HHCE-MOPSO). Furthermore, the feasibility of the hybrid PSO is verified by the simulation of the multi-objective test function. The results not only broaden the new theory and method of clustering algorithm for unbalanced data, but also verify the accuracy and feasibility of the hybrid PSO. Furthermore, the clustering analysis method based on information entropy is a new method. As a result, the research results have both important scientific value and good practical significance.

**Keywords:** unbalanced data; clustering algorithm; high-dimensional PSO algorithm; cloud model; information entropy

## 1. Introduction

There are various high-dimensional unbalanced data in big data applications. In particular, it is very difficult to deal with high-dimensional multi-view data in classic clustering. Cluster analysis is a clustering method of unsupervised learning as the most important branch of data mining technology that has very important scientific value. Simultaneously, the cluster analysis is a multivariate statistical analysis method to cluster samples by maximizing the similarity of samples within a class and minimizing the similarity of samples between classes. Usually, the intra class squared error is minimized to achieve class division. Concurrently, the cluster analysis can also be regarded as an optimization problem and high-dimensional multi-objective clustering optimization is one of the challenges. Therefore, how to efficiently process such massive data and obtain valuable information from it has become the focus of enterprises, scientific research, and other fields. Data is growing explosively, and a large number of high-dimensional data are generated from different fields every day. Aiming at the exponential and high-dimensional problem, a high-dimensional, multi-objective optimization algorithm with additive structure is proposed [1]. How to identify and mine useful information from massive data is particularly important. Multi-objective optimization algorithms have been widely used in many technical industries [2]. However, in many practical applications, data often presents the characteristics of category imbalance and increases the difficulty of the clustering algorithm. It is widely known that data often has the problem of unbalanced distribution of categories in many practical applications. Because the proposed implementation is very high, even when the problem dimension and population number increases, it can be widely used in real-world optimization problems [3]. At present, a lot of work has been carried out on the clustering of class imbalance data, which has achieved good results. However,

there are still some problems that need to be solved urgently. Consequently, how to design a reasonable data balance processing method is a very meaningful problem.

Clustering has very important scientific research value such as being the most important branch of data mining technology. In addition, data imbalance is an urgent problem to be solved in the field of high-dimensional complex data clustering in practical projects [4]. Under the background of highly unbalanced clustering data, some scholars apply generalized clustering bootstrapping to Gaussian quasi likelihood and robust estimation methods [5]. Simultaneously, the critical factor affecting the performance improvement of unbalanced data is the selection of clustering methods. Aiming at the limitations of existing unbalanced data modeling techniques, a hybrid clustering algorithm based on boundary sparse samples is proposed. Eventually, the experimental results show that the algorithm has good performance on different data sets with an average increase of 3.5% [6]. In the meantime, the clustering methods for serious class imbalance data and the clustering performance of these methods will significantly decline with the increase in the imbalance ratio. By extending the nonparametric estimation method, the clustering data can satisfy the similarity of objects in the same cluster. In this way, the objects between different clusters can be reduced [7]. The clustering method is sensitive to the initial solution, and it is easy to fall into the local optimum. Furthermore, a fuzzy c-means clustering algorithm based on optimal selection is proposed [8]. The optimal supervised C-means clustering algorithm discusses the alignment of a supervised orthogonal linear local tangent space [9]. In the meantime, the comparison results show that the local search algorithm is superior to other advanced algorithms in terms of solution quality and running time [10]. Simultaneously, it introduces the high-dimensional particle swarm optimization(PSO) algorithm presented by Kennedy and Eberhart that it is a bio-inspired optimization technique based on arandom search to solve some clustering problems. The above research results further expand the application field of high-dimensional, multi-objective optimization problems and improves the ability of multi-objective PSO to solve practical problems such as pattern recognition, data mining, and machine learning. Therefore, the clustering is not only widely used in real life, but also has great theoretical significance.

## 2. Multi-Dimensional Cloud Model

The cloud model is a kind of uncertainty artificial intelligence theory and method from qualitative to quantitative transformation. At the same time, the cloud model has the characteristics of the fuzzy theory and membership function. Moreover, the cloud model can completely unify the fuzziness and uncertainty of things' cognitive concepts into qualitative and quantitative processing and apply them to data mining, intelligent control, and other fields. In practical applications, we consider that events obey or nearly obey the normal distribution, so we usually use the normal cloud model to analyze problems. Furthermore, we will use $Ex$ to express the expectation of the cloud model because the expectation best represents the average degree and level of all sample points in the cloud model. In addition, we will use $En$ to describe the entropy of the cloud model because the measurement of $En$ is determined by the randomness and fuzziness of the sample points of the cloud model, which can describe the dispersion of cloud droplets in the cloud model. Subsequently, we will use $He$ to represent the super entropy of the cloud model, which can measure the uncertainty measure of the cloud model. The larger $He$ is, the greater the cloud droplet dispersion of the sample point and the larger the range of cloud droplet membership of the cloud model, thus the thicker the sample cloud generated. At the same time, we can find that the application of specific cloud models can be realized through cloud reasoning, cloud computing, cloud clustering, and other methods. Subsequently, the cloud models are transformed from abstract to specific by the cloud digital feature generation cloud droplet process. Through the above analysis, the specific relationship can be defined as follows.

$$\mu(x) = e^{\frac{(x-Ex)^2}{2En'^2}} \tag{1}$$

In the normal cloud model, cloud droplet groups have different contributions to qualitative concepts. In this paper, we use multi-dimensional normal clouds to illustrate the contribution of cloud droplet groups to the concept. It can be concluded that the contribution of all elements of the normal cloud model to the qualitative concept is determined as follows.

$$\Delta C \approx \mu_A(x) * \Delta x / (\sqrt{2\pi}En) \tag{2}$$

Through the above analysis of the normal cloud model, it can be concluded that the total contribution of all elements of the normal cloud model to the qualitative concept is expressed as follows.

$$C = \frac{\int\limits_{-\infty}^{+\infty} \mu_A(x)dx}{\sqrt{2\pi}En} = \frac{\int\limits_{-\infty}^{+\infty} e^{-(x-Ex)^2/(2Ex^2)}dx}{\sqrt{2\pi}En} = 1 \tag{3}$$

The total contribution of all elements in the universe $Ex - 3En, Ex + 3En$ to the concept can be expressed as follows.

$$C_{Ex\pm3En} = \frac{1}{\sqrt{2\pi}En}\int_{Ex-3En}^{Ex+3En} \mu_A(x)dx = 99.74\% \tag{4}$$

The sample cloud drops of cloud model in the interval $[Ex - 2En, Ex + 2En]$ and $[Ex - 3En, Ex + 3En]$ account for about 33.34%. At the same time, the contribution of cloud model sample points to qualitative concepts is about 4.40%. Through the above analysis of the cloud model, the contribution of cloud droplet groups in different cloud model sample areas to qualitative concepts can be described in Figure 1 as follows.

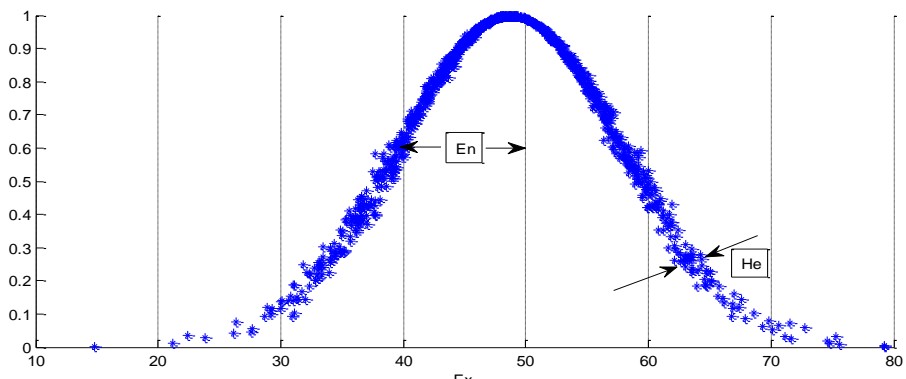

**Figure 1.** The qualitative concept map of the cloud model.

The cloud model is actually a model that transforms qualitative concepts and quantitative data into each other. Positive cloud realizes the transformation from qualitative concept to quantitative data. Through the above analysis, the probability density function expected by the cloud model can be expressed as follows.

$$f_{En'}(x) = \frac{1}{\sqrt{2\pi}He}e^{\frac{(x-En)^2}{2He^2}} \tag{5}$$

The probability density function of the variable can be determined as follows.

$$f_x(x|En') = \frac{1}{\sqrt{2\pi}En'}e^{\frac{(x-Ex)^2}{2En'^2}} \tag{6}$$

where the expectation of variables is $Ex$ and the normal distribution of variance is $En'$. From the conditional probability density formula, it can be inferred that the probability density function can be expressed as follows.

$$f_x(x) = f_{En'}(x) \times f_x(x|En') = \int_{-\infty}^{+\infty} \frac{1}{2\pi He|y|} e^{\frac{(x-Ex)^2}{2y^2} - \frac{(y-En)^2}{2He^2}} dy \tag{7}$$

For any variable, the corresponding function value can be obtained through numerical integration. When the number of cloud drops is cloud drops, the time parzen method can be used to estimate the probability density function. Consequently, the probability density function of the cloud model at that time $He = 0$ can be expressed as follows.

$$f(x) = \frac{1}{\sqrt{2\pi}En} e^{\frac{(x-Ex)^2}{2En^2}} \tag{8}$$

Because all cloud droplets come from the expectation that they are normal random variables, the expectation and variance have the conclusions $EX = Ex$ and $DX = En^2 + He^2$. It is widely known that the traditional algorithms are easy to fall into local optimum and the convergence speed is slow. In order to overcome these shortcomings, a particle swarm optimization algorithm based on the cloud model is proposed. Simultaneously, we can assume that the two-dimensional cloud is represented by $C_{i1} = (Ex_{i1}, En_{i1}, He_{i1})$ and $C_{i2} = (Ex_{i2}, En_{i2}, He_{i2})$. Through the above analysis, the shape similarity calculation based on normal two-dimensional cloud is shown as follows.

$$Sim_{si}(C_{i1}, C_{i2}) = \frac{\min\left(\sqrt{En_{i1}^2 + He_{i1}^2}, \sqrt{En_{i2}^2 + He_{i2}^2}\right)}{\max\left(\sqrt{En_{i1}^2 + He_{i1}^2}, \sqrt{En_{i2}^2 + He_{i2}^2}\right)} \tag{9}$$

Through the above analysis, the distance similarity based on the normal cloud model can be calculated as follows.

$$Sim_{di} = \frac{1}{\sqrt{(Ex_{i1} - Ex_{i2})^2 + (En_{i1} - En_{i2})^2 + (He_{i1} - He_{i2})^2}} \tag{10}$$

Therefore, the comprehensive similarity calculation of two-dimensional cloud model can be expressed as follows.

$$\mu_i = Sim_{ci} = Sim_{si} \times Sim_{di} \tag{11}$$

Multi-dimensional data refers to a data set composed of a large number of samples with multi-dimensional attributes. Concurrently, the multi-dimensional cloud model is a cloud model generated according to various characteristics of data to establish an integrated cloud. Furthermore, the system can be evaluated by calculating the similarity between the multi-dimensional cloud model composed of new data and the integrated cloud. As a result, the multi-dimensional cloud model can effectively solve the problem of low clustering accuracy caused by the fuzziness and uncertainty of multi-sample, multi-dimensional data. In the meantime, the simulation results of the multi-dimensional cloud model can be represented in Figure 2 as follows.

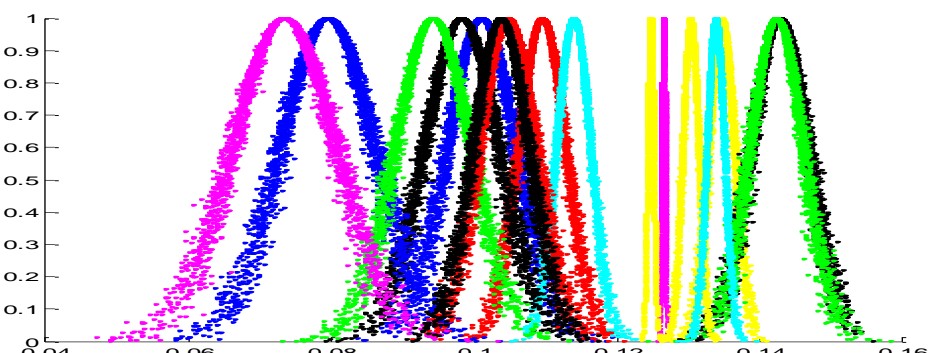

**Figure 2.** The simulation diagram of the multi-dimensional cloud model.

Due to the uncertainty and multidimensional attributes of complex data, it is difficult to obtain high clustering accuracy by directly using existing machine learning algorithms and statistical analysis methods. At the same time, it can be found, through experiments, that this clustering should have computational efficiency and be completed in the shortest time. In order to solve the two problems of similarity and attribute reduction, we propose a high-dimensional, multi-objective particle swarm optimization hybrid clustering algorithm based on the cloud model and entropy.

## 3. HybridHigh-Dimensional, Multi-Objective PSO Clustering Algorithm Based on the Cloud Model and Entropy

Clustering is a critical technology and method in data mining, machine learning, pattern recognition, and artificial intelligence. Through a clustering algorithm, we can find the global distribution pattern and the relationship between data attributes. In particular, the clustering algorithm has the ability of scalability, processing different types of data, and processing high-dimensional data. Therefore, the emergence of large-scale unbalanced data sets poses special challenges to clustering analysis technology. In the meantime, the hybrid PSO algorithm has a global optimization ability and distributed random search characteristics to solve the problems that traditional clustering algorithms easily fall into, including local optimization and sensitive to initial value [11,12]. Simultaneously, the convergence speed and global optimization ability of the algorithm can be effectively improved through the combination of PSO, which combines excellent features and traditional clustering methods [13–19]. In order to effectively reduce the algorithm from falling into the local optimum, the multi-mode cooperative multi-objective particle swarm optimization algorithm based on reinforcement learning is proposed. Furthermore, the experimental results show that the hybrid PSO algorithm is more effective and robust than the other algorithms [20]. As a powerful optimization technology, a feature selection clustering method based on particle swarm optimization (PSO) algorithm is proposed [21]. The particle swarm optimization (PSO), based on multi subgroup distributed architecture, is very effective for static multi-objective optimization problems, but it has not been used to solve dynamic multi-objective problems (DMOP) at present [22]. According to the multi-objective particle swarm optimization (MOPSO) algorithm, the design of the updating mechanism and population maintenance mechanism is the critical technology to obtain the optimal solution [23]. In order to effectively solve multi-modal, multi-objective optimization problems with the same fitness value, an input-output fuzzy clustering was proposed [24]. Based on the feature selection algorithm, a new hybrid clustering algorithm for multi-objective PSO feature selection is proposed [25]. Particle swarm optimization has a built-in guidance strategy to improve their solutions [26]. Combined with the automatic clustering problem, a hybrid method of chaos game optimization and particle swarm optimization (CGOPSO) is proposed [27]. Simultaneously, the proposed clustering algorithm is compared with spherical K-means and the PSO algorithm in terms of feasibility and convergence characteristics [28]. However, it is easy to converge to the suboptimal

clustering solution too early and the learning coefficient value needs to be adjusted to find a better solution [29]. Therefore, the particle swarm optimization (PSO) algorithm is widely used in clustering analysis.

By optimizing the algorithm to find the best cluster center, the hybrid ensemble clustering algorithm is used as an optimization problem to solve [30]. At the same time, integrated clustering has been used as the final clustering to achieve high quality. On the basis of data analysis, it is a good idea to improve the performance and the speed in the field of machine learning [31]. At the same time, we all know that the fundamental reason for the decline of clustering performance of imbalanced sets lies in the high-dimensional characteristics of imbalanced data. However, optimization problems show a trend of diversification accompanied by non-linear, high dimensional, and other characteristics. The optimization objectives involved in these practical problems may be one or even several conflicting ones, often with harsh constraints. The PSO algorithm is an intelligent optimization with few adjustable parameters and good robustness. With its outstanding performance in dealing with single objective optimization problems, it has been extended to the field of multi-objective optimization. Through the comparison with the comparison algorithm, it shows that the superiority of the PSO depends on the parameter tuning and it has premature convergence. The clustering evaluation function of data set is regarded as the fitness function of PSO algorithm, which it is a well-known, unsupervised clustering algorithm. As a result, the possible clustering partition is regarded as a particle of the population, so that the hybrid PSO can realize the function of the optimal clustering partition and search efficiently and escape from local optima.

On the basis of the above analysis, we constructed a fitness function similar to the multi-objective PSO algorithm based on the objective function and constraints of the multi-objective optimization problem. In the given multi-objective interval PSO algorithm, initial conditions and fitness function particle search space that are optimal variables and encoded as the position. During the iteration of multi-objective interval particle swarm optimization algorithm, particles track their own optimal position (*pbest*) and the group optimal position (*gbest*). Consequently, the tracking equation of multi-objective PSO algorithm can be expressed as follows.

$$\begin{cases} v_{i_d}^{k+1} = v_{i_d}^{k} + c_1 r_1 (pbest_{i_d}^{k} - x_{i_d}^{k}) + c_2 r_2 (gbest_{i_d}^{k} - x_{i_d}^{k}) \\ x_{i_d}^{k+1} = x_{i_d}^{k} + t_0 * v_{i_d}^{k+1} (t_0 = 1) \end{cases} \tag{12}$$

Among them, $c_1$ and $c_2$ are the acceleration constants for multi-objective interval PSO algorithm to adjust the maximum step size of particles to reach the optimal position (*pbest*) and the optimal position (*gbest*). Where $x_{i_d}^{k+1} = x_{i_d}^{k} + v_{i_d}^{k+1}$. Each particle is assigned an initial position and initial velocity as the optimal initial state. $r_1(x)$ and $r_2(x)$ are uniformly distributed random numbers to simulate the group behavior of the multi-objective interval PSO algorithm. In order to improve the convergence speed and solution quality of multi-objective interval particle swarm optimization algorithm, the basic PSO iterative equation is improved using the following expression.

$$v_{i_d}^{k+1} = w v_{i_d}^{k} + c_1 r_1 (pbest_{i_d}^{k} - x_{i_d}^{k}) + c_2 r_2 (gbest_{i_d}^{k} - x_{i_d}^{k}) \tag{13}$$

Along with these parameters, $w$ is the inertial weight coefficient. Larger inertia weight can enhance the global search ability of the multi-objective particle swarm optimization algorithm. With the reduction of inertia weight, the multi-objective PSO algorithm realizes the local search and finally gets the optimal solution. Finally, we propose to introduce a compression factor into the multi-objective PSO algorithm to ensure the convergence equation of the algorithm.

Through the above analysis, the convergence equation of the algorithm can be expressed as follows.

$$v_{i_d}^{k+1} = \chi \left\{ v_{i_d}^k + c_1 r_1 (pbest_{i_d}^k - x_{i_d}^k) + c_2 r_2 (gbest_{i_d}^k - x_{i_d}^k) \right\} \tag{14}$$

where $(\varphi = c_1 + c_2)$ is the compression factor of the multi-objective particle swarm optimization algorithm. $\chi = 2 / \left| 2 - \varphi - \sqrt{\varphi^2 - 4\varphi} \right|$.

Through research of the PSO and multi-objective PSO algorithm, we can conclude that the average focusing distance of the algorithm particles $\tilde{d}$, the maximum focusing distance between particles $d_{\max}$, and the change rate of the particle focusing on distance $K$ are shown as follows.

$$\begin{cases} \tilde{d} = \dfrac{\sum\limits_{i=1}^{n} \sqrt{\sum\limits_{i=1}^{m} (p_{gm} - x_{id})^2}}{n} \\ d_{\max} = \max \left( \sqrt{\sum\limits_{i=1}^{m} (p_{gm} - x_{id})^2} \right) \end{cases} \tag{15}$$

$$K = \dfrac{\left( \max\left( \sqrt{\sum\limits_{i=1}^{m} (p_{gm} - x_{id})^2} \right) - \dfrac{\sum\limits_{i=1}^{n} \sqrt{\sum\limits_{i=1}^{m} (p_{gm} - x_{id})^2}}{n} \right)}{\max\left( \sqrt{\sum\limits_{i=1}^{m} (p_{gm} - x_{id})^2} \right)} \tag{16}$$

Through in-depth research on the multi-objective PSO algorithm, the following functions are used to explore the properties and related mechanisms of the multi-objective PSO algorithm and to perform the following simulation experiments. The simulation results of its multi-objective function can be shown in Figure 3 as follows.

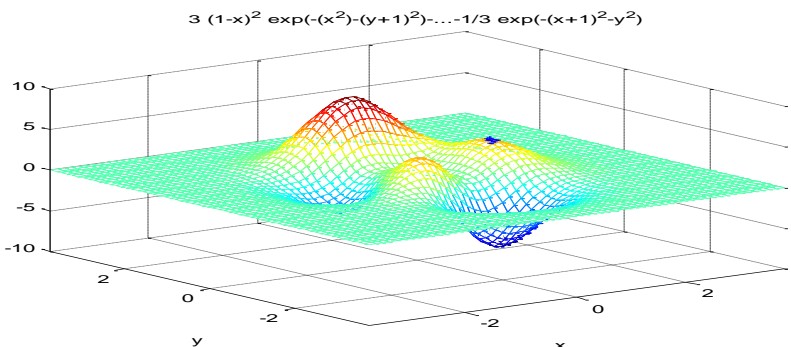

**Figure 3.** The simulation results of multi-objective function.

In the multi-objective PSO algorithm, it is often difficult to get rid of the local optimum, the accuracy of the solution is not enough, the calculation speed is not fast enough, and so on. As the number of problem objectives increases, the dimension and scale of the solution increase accordingly, resulting in high complexity of the problem and greater difficulty in solving it. How to effectively solve the quality and efficiency of complex optimization problems based on swarm intelligence algorithms has become a hotspot in this field. For multi-objective optimization problems, we should not only solve the balance between diversity and convergence, but also consider the optimal particle selection strategy and diversity maintenance mechanism and other issues effectively. Through a series of experiments, the convergence of different PSO algorithms with different particle sizes can be represented in Figure 4 as follows.

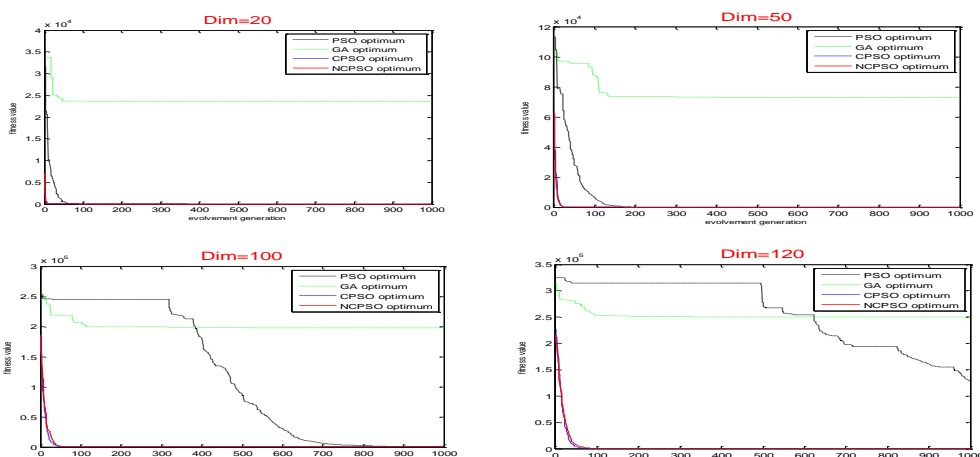

**Figure 4.** The convergence of different algorithms with different sizes.

Through experiments, we found that the multi-objective PSO algorithm has the advantages of a small number of individuals, fast iterative convergence, simple operation, and easy implementation. At the same time, we found that the PSO is a random search algorithm based on probability, which has strong robustness and global optimization ability. In the process of generating initial particles, the strategy makes full use of the correlation between features to make the initial population more competitive. Therefore, the multi-objective PSO algorithm has a wide range of applications. However, when dealing with feature selection of high-dimensional data, most of the existing feature selection methods based on the PSO algorithm are prone to fall into local optimization, high computational cost, premature convergence, and low search efficiency. Some scholars also proposed intelligent clustering technology to improve the convergence performance of clustering technology [16]. It is very difficult for classical clustering algorithms to process high-dimensional data. The hybrid algorithms should consider all features of the data and the correlation of all features.

The PSO algorithm guides particles to move in the search space according to the individual optimal position and the global optimal position. Concurrently, this strategy is simple and efficient, which easily leads to the problem of particle oscillation in the search process. Thus, the search efficiency of the particle swarm optimization algorithm is reduced and some solutions with better performance are missed. Furthermore, the core idea of the hybrid clustering algorithm is to generate a large number of high-quality feature subsets using the correlation information of features. It not only has a good clustering effect, but also has good dimension reduction efficiency when facing unbalanced data. Thus, it can be applied to the clustering problem of unbalanced data. Due to the unordered range of clustering data, it is impossible to compare the numerical values. The distance-based measurement method cannot be used to measure the similarity between objects. Simultaneously, we will study the clustering algorithm based on information entropy for the above clustering data. Based on the PSO algorithm, the cloud model is applied to deal with uncertainty with the idea of information entropy.

The lowest value of information entropy is selected as the classification class, and the class with higher information entropy is used as the next target object, and so on, until a cluster that matches the k value is found. Furthermore, the idea of this clustering algorithm is based on the theory of information entropy, and the average information gain rate is proposed according to the relevant knowledge of information entropy. The highest value of mean information gain rate is selected as the standard of equivalence classification. In addition, we introduce information entropy and mutual information to measure the importance of features and select features with high information entropy. Subsequently, the data can be mapped from a high-dimensional feature space to a low dimensional space

through information entropy features. As a result, the joint entropy of two variable random variables can be expressed as follows.

$$H(X, Y) = E\left[\log \frac{1}{p(x, y)}\right] = \sum_x \sum_y p(x, y) \log \frac{1}{p(x, y)} \tag{17}$$

Under the condition that random variable $X$ is determined, the conditional entropy of random variable $Y$ can be expressed as follows.

$$H(Y|X) = \sum_x \sum_y p(x, y) \log \frac{1}{p(y|x)} = \sum_x p(x) \sum_y p(y|x) \log \frac{1}{p(y|x)} \tag{18}$$

Through the above analysis, it can be expressed as follows.

$$H(Y|X) = \sum_x p(x) H(Y|x) = E\left[\log \frac{1}{p(y|x)}\right] \tag{19}$$

To solve the above problems, we propose a hybrid, high-dimensional, multi-objective PSO clustering algorithm based on the multi-dimensional cloud model and entropy theory(HHCE-MOPSO) and the feature selection algorithm based on correlation information entropy and the particle swarm optimization algorithm. Simultaneously, the particle position vector is used as the attribute weight vector and the information entropy is used as the attribute weight evaluation function. In addition, the gradient descent method is used to minimize the attribute weight evaluation function. In the process of clustering, the influence of intra class entropy and inter class entropy on attribute weight is comprehensively considered and a group of optimal attribute weight values are finally obtained through iteration. Thus, the global search ability and convergence speed of the algorithm are improved. In the meantime, the hybrid method firstly uses correlation-based information entropy to reduce the high dimension of features quickly. Then, the optimization features are searched by the PSO algorithm. Similar or redundant features are divided into one feature class by information entropy to reduce the search time and space. The particle swarm is prevented from falling into local optimization to obtain the optimal solution of the final feature set. Finally, we will consider large-scale, high-dimensional data that is difficult to process by existing particle swarm feature selection algorithms and propose anovel hybrid, high-dimensional, multi-objective PSO clustering algorithm based on the cloud model and information entropy.

## 4. Numerical Example

In the multi-objective particle swarm optimization algorithm, particles can be selected based on the Pareto optimal individual extremum. The current position of its particles are compared with the best position in history. Finally, a satisfactory solution, which is consistent with the actual situation, is selected as the individual extremum of the particle. At the same time, we apply the following four experimental functions to do simulation experiments. $F1 = -(x_1 - 0.5)^4 - (x_2 - 0.2)^4 + 2$. $F2 = -0.5 + (\sin \sqrt{x_1{}^2 - x_2{}^4})^4 - 0.5/(1 + 0.01(x_1{}^2 + x_2{}^4))^4$. $F3 = -(x_2 - 0.5)^4 - (x_1 \sin(x_2 - 0.5)$. $F4 = -(x_1 - 0.7)^4 - 10\cos(2\pi x_2) + 10$. Through testing different functions, the experimental results of the multi-objective function based on the multi-objective PSO algorithm are shown in Table 1 as follows.

**Table 1.** Function experiment results based on multi-objective PSO.

| N | F1 | F2 | F3 | F4 |
|---|---|---|---|---|
| 1 | 0.106499808405461 | 0.067330182240415 | 0.124377116301462 | 0.499955546962077 |
| 2 | 0.049737153906336 | 0.086820395586198 | 0.152198389566321 | 0.498713173701964 |
| 3 | 0.012271391505579 | 0.063407876573236 | 0.118557185891584 | 0.500789090951651 |
| 4 | 0.059748779007947 | 0.079984601159047 | 0.181778700051687 | 0.500001701392795 |
| 5 | 0.007182350707380 | 0.044994628705243 | 0.106369496986030 | 0.496933444858318 |
| 6 | 0.028311613137651 | 0.074090886988584 | 0.275260280239408 | 0.498887957603687 |
| 7 | 0.035276749714561 | 0.068119108505511 | 0.174298597155842 | 0.498458132620899 |
| 8 | 0.018398812996123 | 0.059303292690439 | 0.164149889811481 | 0.498878446602397 |
| 9 | 0.017724730373242 | 0.061147877561301 | 0.229416799984821 | 0.498599468093871 |
| 10 | 0.087259472853428 | 0.110801090927873 | 0.268141848002487 | 0.500060922016868 |
| 11 | 0.025138533599246 | 0.052033696500143 | 0.296139928492164 | 0.499667735329447 |
| 12 | 0.015732190370921 | 0.074778835122926 | 0.210926317455325 | 0.500978252622088 |
| 13 | 0.053251800555326 | 0.090238137048483 | 0.237727894208217 | 0.499292970499502 |
| 14 | 0.006365915863766 | 0.032023220957725 | 0.263193322717373 | 0.501448338212031 |
| 15 | 0.019652879857307 | 0.053962701317858 | 0.334056520900436 | 0.497959864836247 |
| 16 | 0.048206842506823 | 0.063330248205566 | 0.295396187276877 | 0.500654771790587 |
| 17 | 0.069439335270074 | 0.044399734565123 | 0.265652503713111 | 0.500439821847981 |
| 18 | 0.013099803987706 | 0.057949862222682 | 0.244600657928145 | 0.499580800950720 |
| 19 | 0.014416740076318 | 0.056339654451476 | 0.266422324322419 | 0.500250196137362 |
| 20 | 0.012117224333988 | 0.052303264496226 | 0.305532364272770 | 0.499126078903273 |
| 21 | 0.042866757575654 | 0.089425849796620 | 0.312086205207292 | 0.500010829781512 |
| 22 | 0.014647138074965 | 0.058134128590683 | −0.056408677384264 | 0.498794765432105 |
| 23 | 0.088041986812975 | 0.102084985512765 | 0.205532364272770 | 0.498975149753489 |
| 24 | 0.090890557660351 | 0.095408489660995 | 0.179540992049498 | 0.498666077341443 |
| 25 | 0.015508456658759 | 0.064955261142703 | 0.279540992049498 | 0.498220217186682 |
| 26 | 0.022130504550377 | 0.075503679089671 | 0.060569639705176 | 0.499393068016345 |
| 27 | 0.083906502098294 | 0.085684293312267 | 0.249392149808258 | 0.500079250249175 |
| 28 | 0.027295599615759 | 0.065948187669335 | 0.216639604236573 | 0.500225643158520 |
| 29 | 0.007686972446238 | 0.034829934566585 | 0.144233559593972 | 0.499288863741454 |
| 30 | −0.000087959115083 | −0.004312468629715 | 0.261281758581385 | 0.499781973533050 |
| 31 | 0.031525187898891 | 0.067696065331331 | 0.158604428509587 | 0.499187353748357 |
| 32 | 0.043495161666248 | 0.080722762768860 | −0.008594010632467 | 0.498532431799683 |
| 33 | 0.066343902124794 | 0.069243776791827 | 0.341841583248901 | 0.499840866124779 |
| 34 | 0.102273774248894 | 0.095695989072027 | 0.252887495462082 | 0.499923364181269 |
| 35 | 0.097427741097806 | 0.098420780466412 | 0.186682617704863 | 0.499829479413832 |
| 36 | 0.110900310094521 | 0.085422503540809 | 0.336538863373088 | 0.500550587358501 |
| 37 | 0.063572037190324 | 0.061386120975582 | 0.291483641237078 | 0.500164235318542 |
| 38 | 0.163896033773297 | 0.088045190537832 | 0.278063228327557 | 0.497931277694565 |
| 39 | 0.076041876860540 | 0.086341096182584 | 0.366426024723916 | 0.499321870487254 |
| 40 | 0.036631149052383 | 0.030406015500888 | 0.343939793326859 | 0.500131940026728 |
| 41 | 0.153224698735289 | 0.105774572764509 | 0.285276499719910 | 0.499966015358706 |
| 42 | 0.036631149052383 | 0.030406015500888 | 0.287189676986890 | 0.499769791524706 |
| 43 | 0.059736827073072 | 0.067027874851685 | 0.347925259918005 | 0.499587528992349 |
| 44 | 0.022869907090576 | 0.062205089212256 | 0.387189676986890 | 0.499273538602641 |
| 45 | 0.026102441960279 | 0.069621787371245 | 0.414726237619834 | 0.500619836407797 |
| 46 | 0.004437754638153 | 0.020990335771239 | 0.484189676986890 | 0.498792173268238 |
| 47 | 0.115122646687255 | 0.081766041434314 | 0.504618724389848 | 0.499304770335719 |
| 48 | 0.074167751678719 | 0.071559016333562 | 0.404618724389848 | 0.499907527996732 |
| 49 | 0.009090076889129 | 0.050091362287326 | 0.412792112794205 | 0.498914307071991 |
| 50 | 0.057916908344494 | 0.067453349159717 | 0.406802282755005 | 0.499529080847996 |

The results show that the hybrid, high-dimensional, multi-objective PSO clustering algorithm is very efficient in terms of creating well separated, compact, and sustainable clusters in Table 1. Then, the multi-objective PSO algorithm is applied to conduct Pareto vector simulation experiments on the above multi-objective functions. The experimental results are shown in Figure 5 as follows.

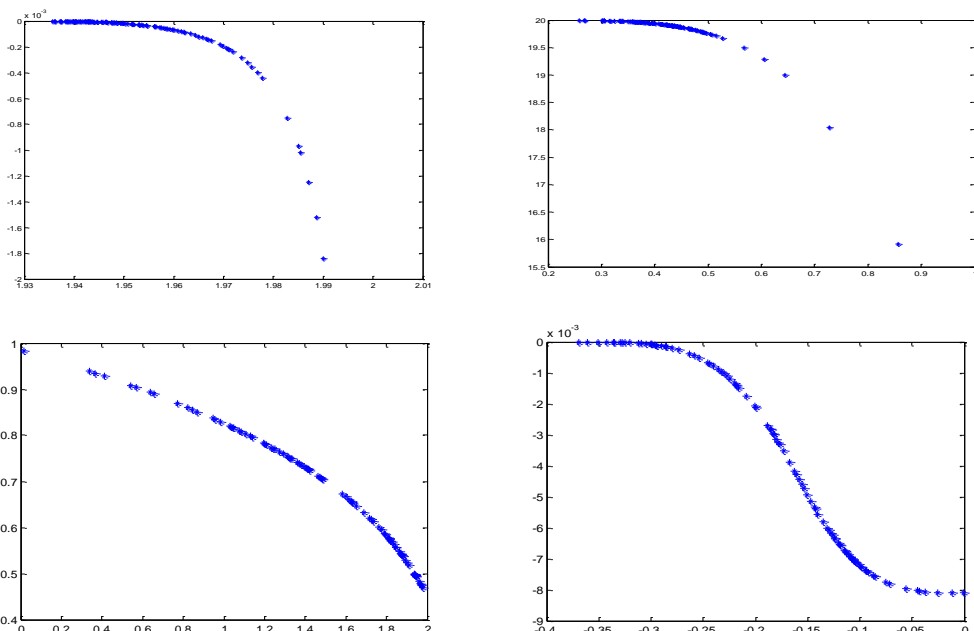

**Figure 5.** The Pareto optimal solution of test function.

From the above analysis, it can be concluded that the HHCE-MOPSO algorithm has better performance than the optimization algorithm, which provides a scientific experimental basis for the future numerical experiments. Metaheuristic and unsupervised PSO clustering algorithms require tuning parameters that control the scope of exploration and exploitation during the optimal search in search space. From Figure 5, it can be observed that the parameter settings are crucial for the hybrid algorithms to function effectively and can significantly affect the performance results.

In order to verify the feasibility of the hybrid PSO algorithms, we applied the following functions to carry out a series of simulation experiments. Based on the above analysis, we used the same test function such as Rosenbrock and different hybrid PSO algorithms. In addition, the experimental results with the best performance of the same test function are described in Figures 6–10 as follows.

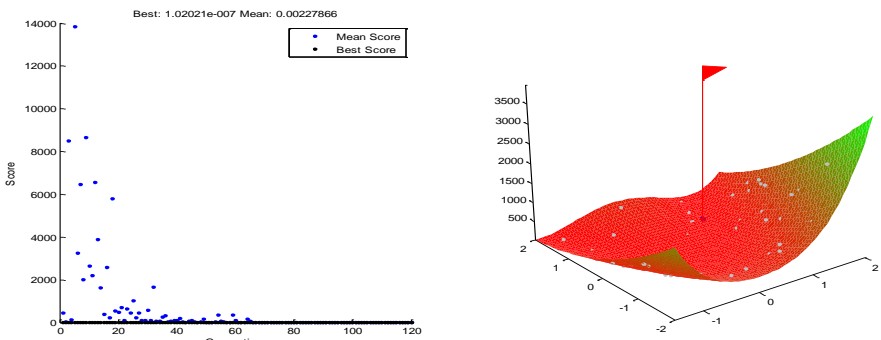

**Figure 6.** The experimental results of Rosenbrock by HHCE-MOPSO.

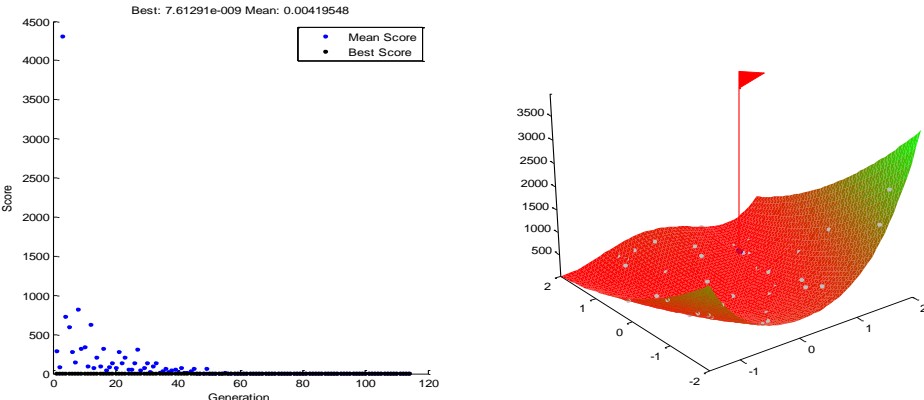

**Figure 7.** The experimental results of Rosenbrock by C-MOPSO.

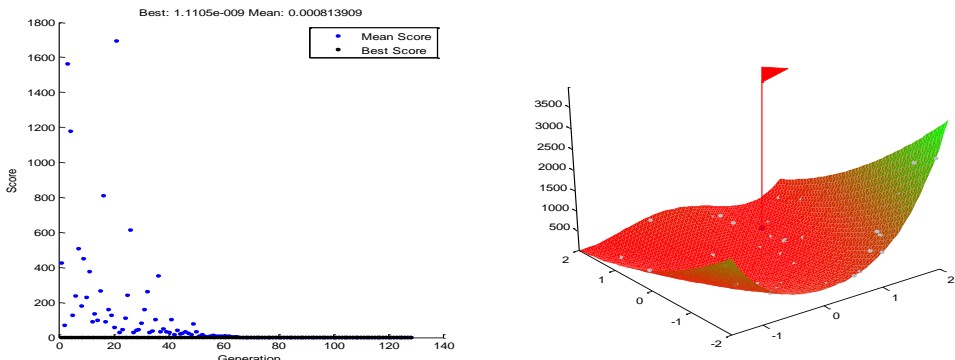

**Figure 8.** The experimental results of Rosenbrock by E-MOPSO.

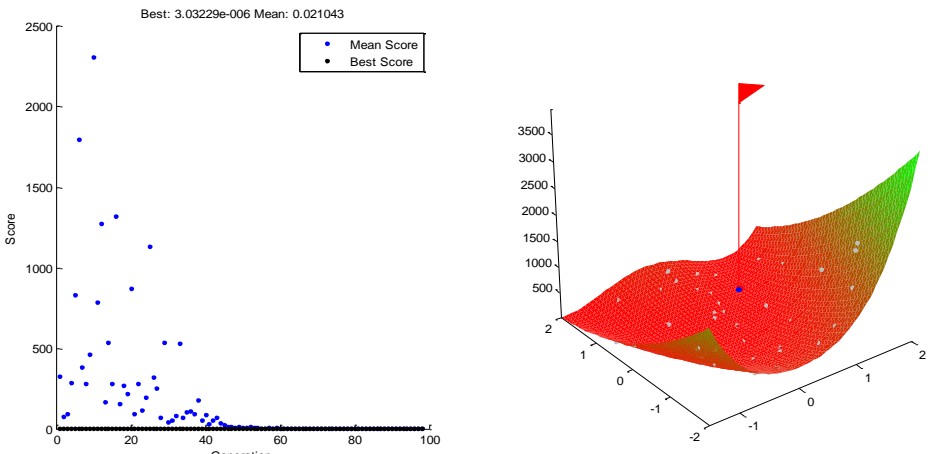

**Figure 9.** The experimental results of Rosenbrock by MOPSO.

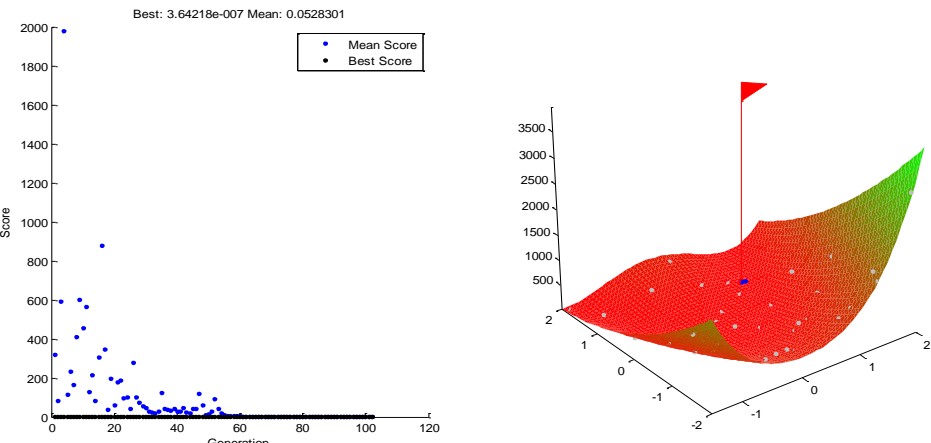

**Figure 10.** The experimental results of Rosenbrock by PSO.

Based on the above analysis, we used the same test function such as Ackleysfcn and different hybrid PSO algorithms. The experimental results of the same test function are described in Figures 11–15 as follows.

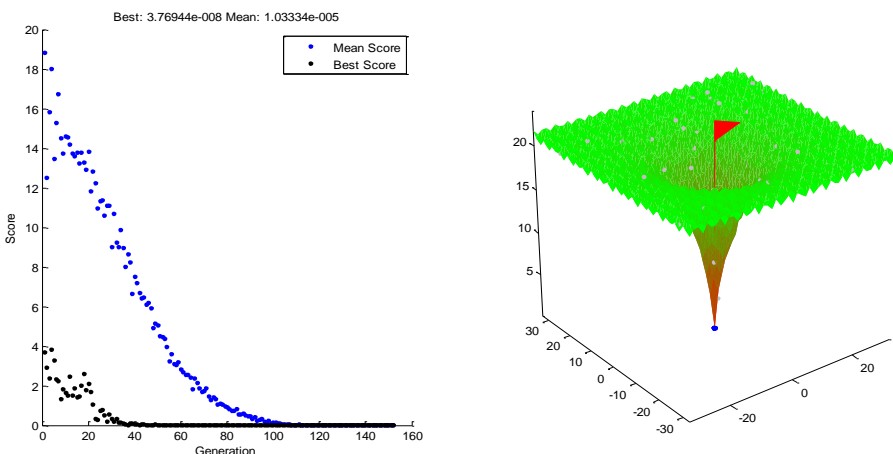

**Figure 11.** The experimental results of Ackleysfcn by HHCE-MOPSO.

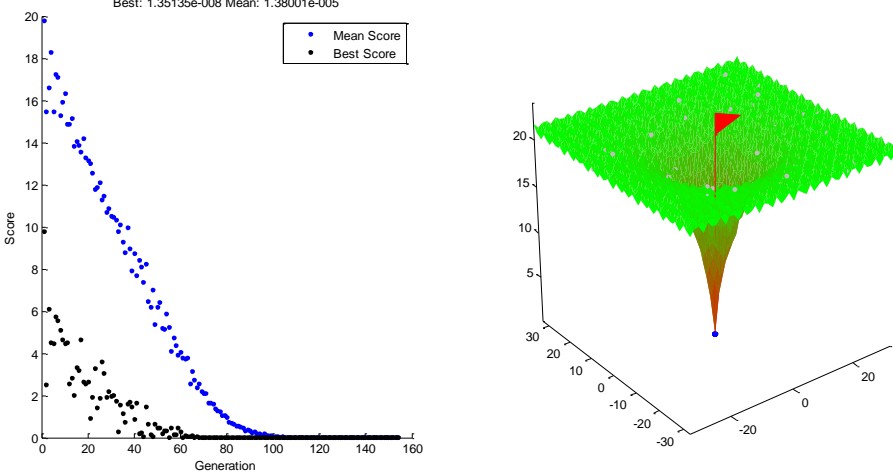

**Figure 12.** The experimental results of Ackleysfcn by C-MOPSO.

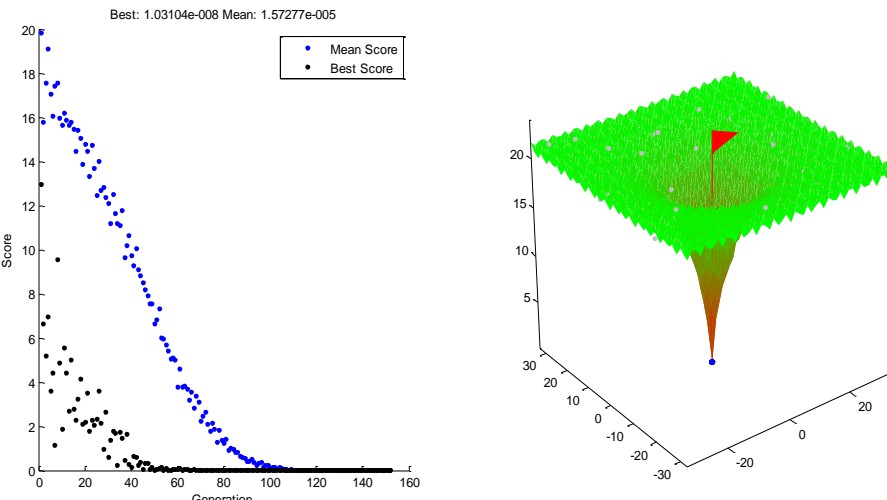

**Figure 13.** The experimental results of Ackleysfcn by E-MOPSO.

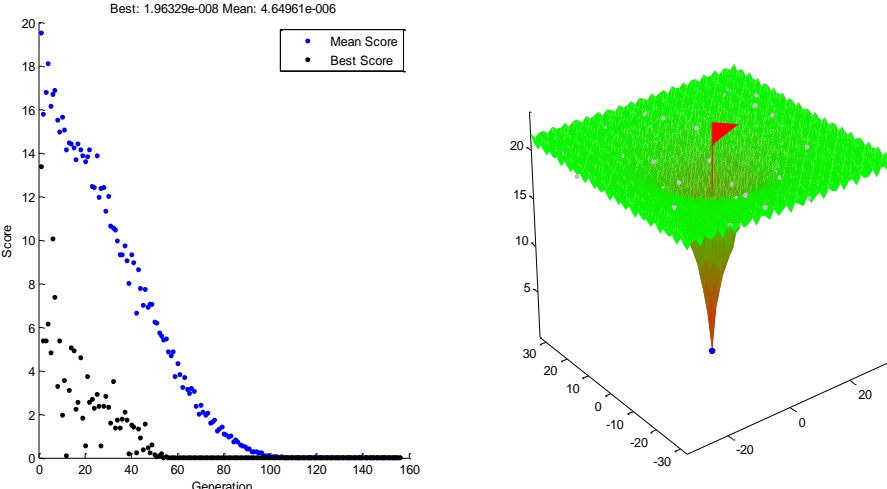

**Figure 14.** The experimental results of Ackleysfcn by MOPSO.

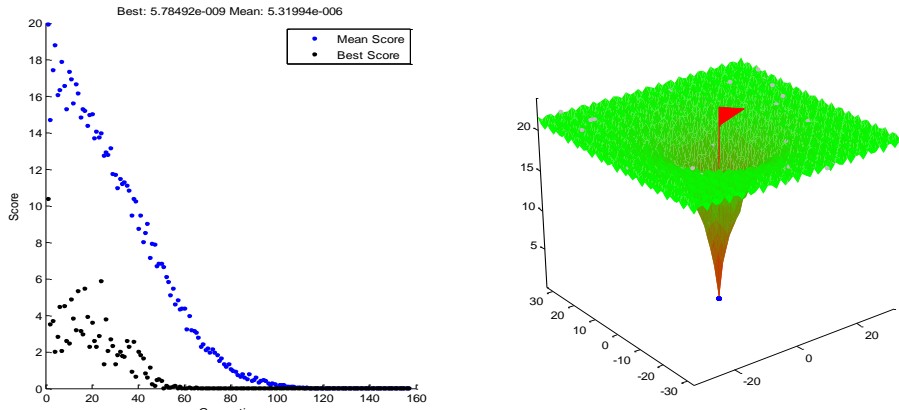

**Figure 15.** The experimental results of Ackleysfcn by PSO.

For quantitative comparison of clustering performance, we used the same test function such as Dejongsfcn and different hybrid PSO algorithms. The experimental results of the same test function are described in Figures 16–20 as follows.

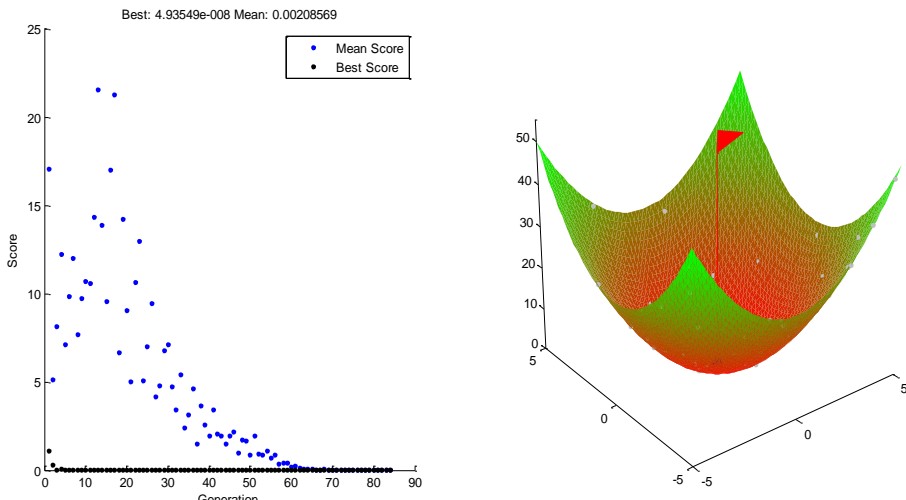

**Figure 16.** The experimental results of Dejongsfcn by HHCE-MOPSO.

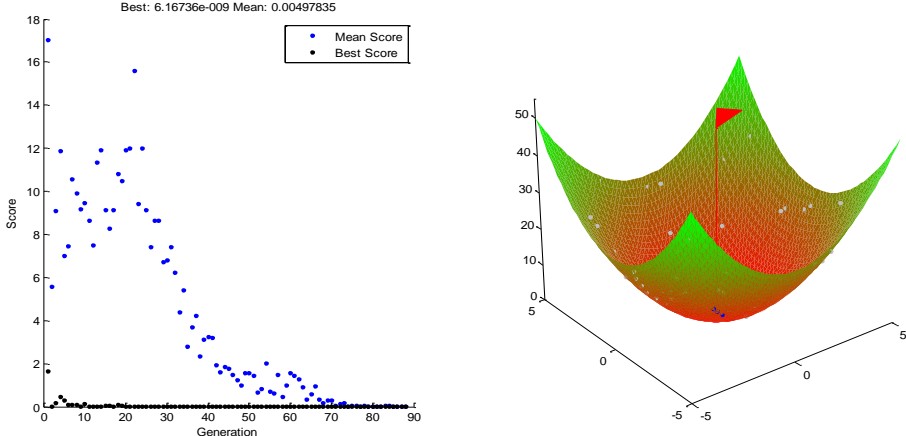

**Figure 17.** The experimental results of Dejongsfcn by C-MOPSO.

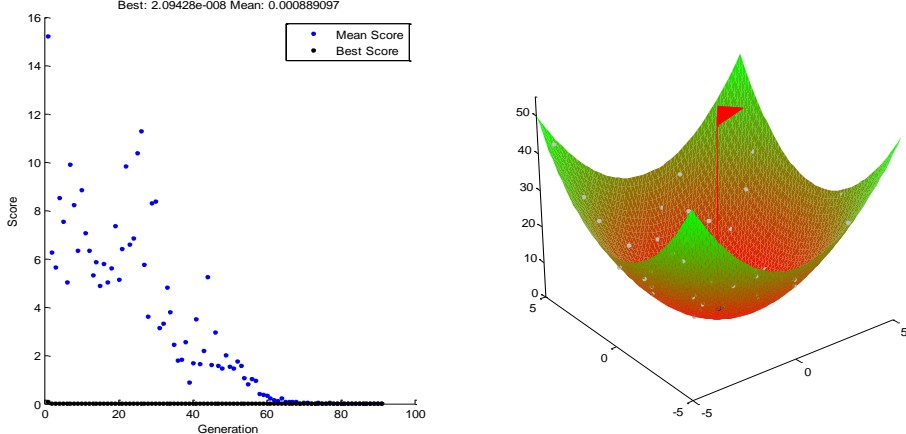

**Figure 18.** The experimental results of Dejongsfcn by E-MOPSO.

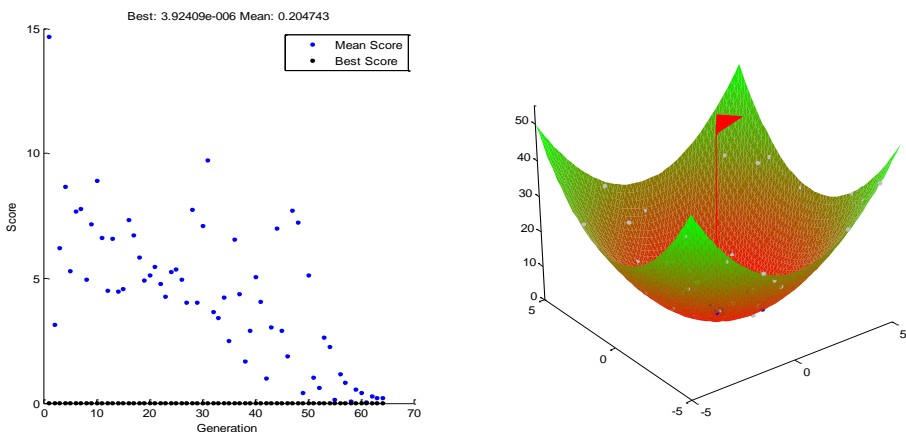

**Figure 19.** The experimental results of Dejongsfcn by MOPSO.

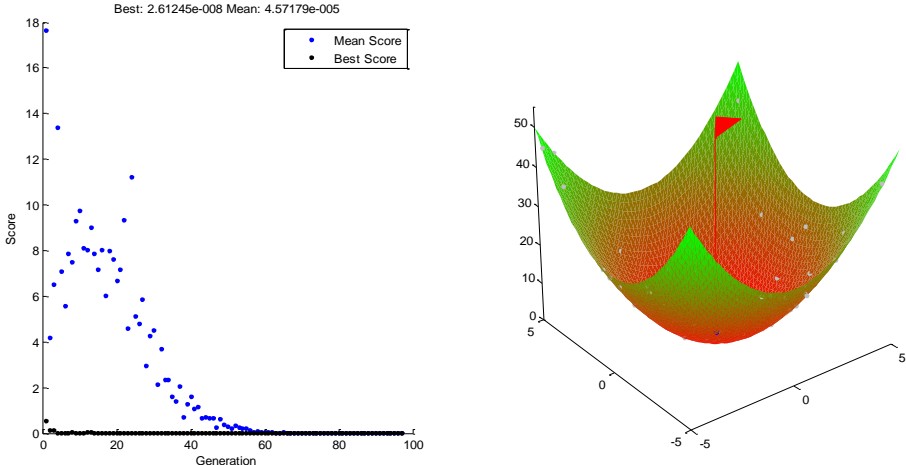

**Figure 20.** The experimental results of Dejongsfcn by PSO.

Based on the above analysis, we will use the same test function such as Dropwavefcn and different hybrid PSO algorithms. In the meantime, the experimental results of the same test function are described in Figures 21–25 as follows.

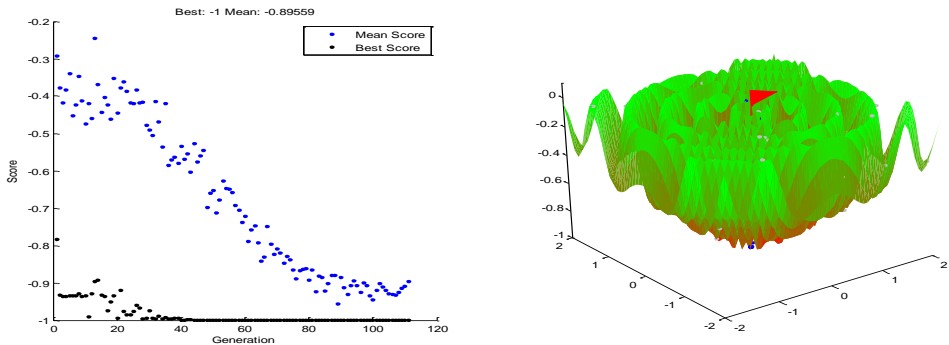

**Figure 21.** The experimental results of Dropwavefcn by HHCE-MOPSO.

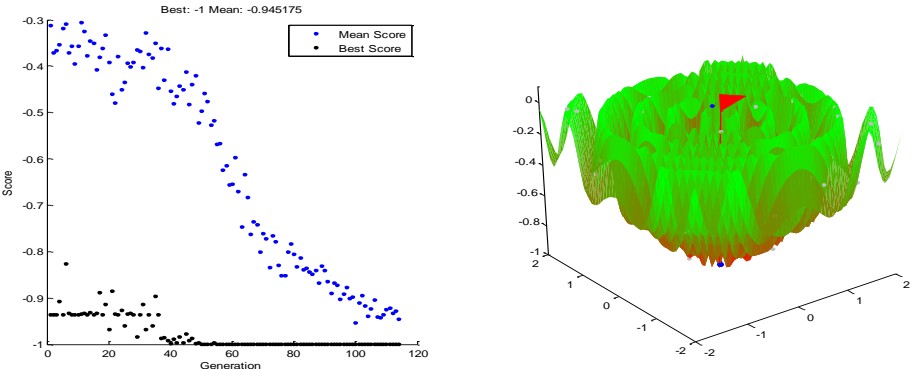

**Figure 22.** The experimental results of Dropwavefcn by C-MOPSO.

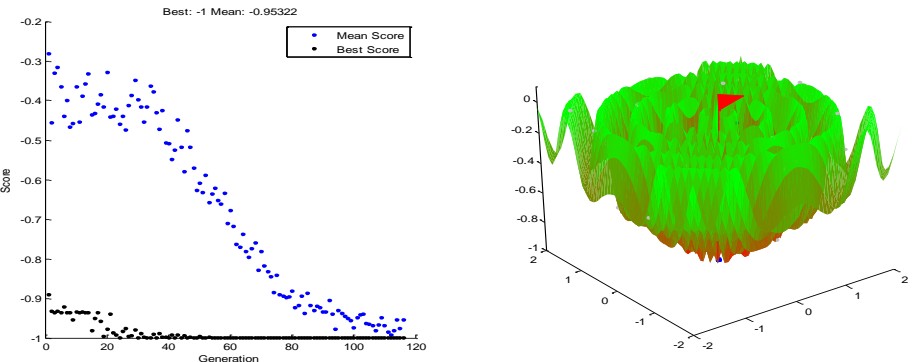

**Figure 23.** The experimental results of Dropwavefcn by E-MOPSO.

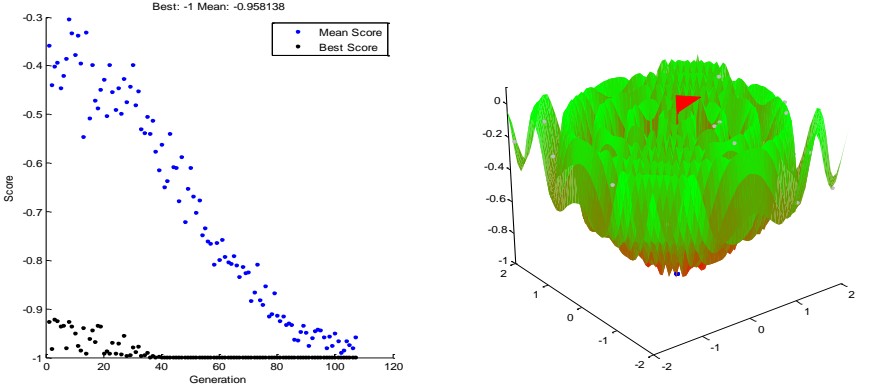

**Figure 24.** The experimental results of Dropwavefcn by MOPSO.

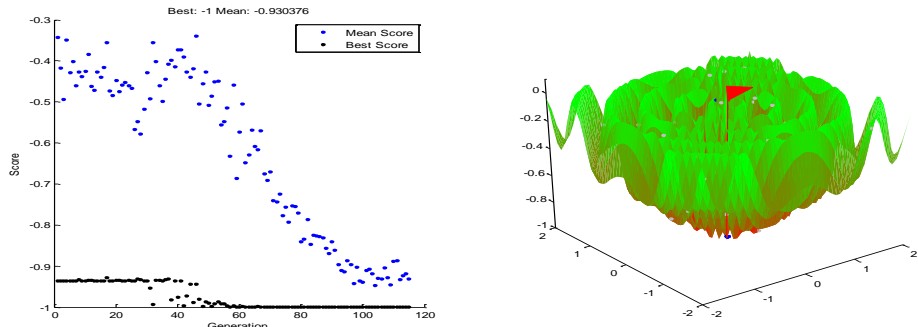

**Figure 25.** The experimental results of Dropwavefcn by PSO.

For quantitative comparison of clustering performance, we used the same test function such as Griewangksfcn and different hybrid PSO algorithms. The experimental results of the test function are described in Figures 26–30 as follows.

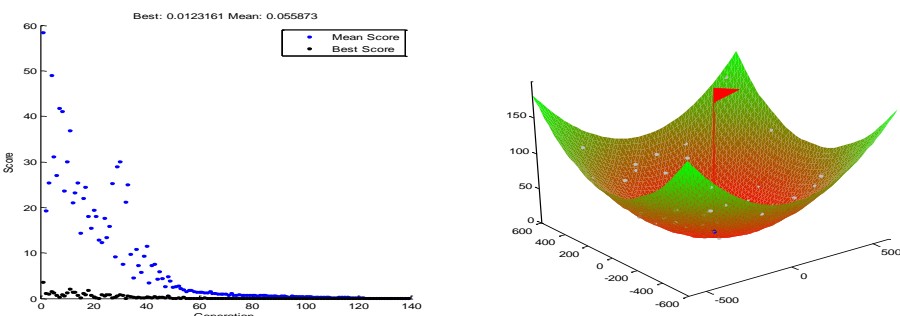

**Figure 26.** The experimental results of Griewangksfcn by HHCE-MOPSO.

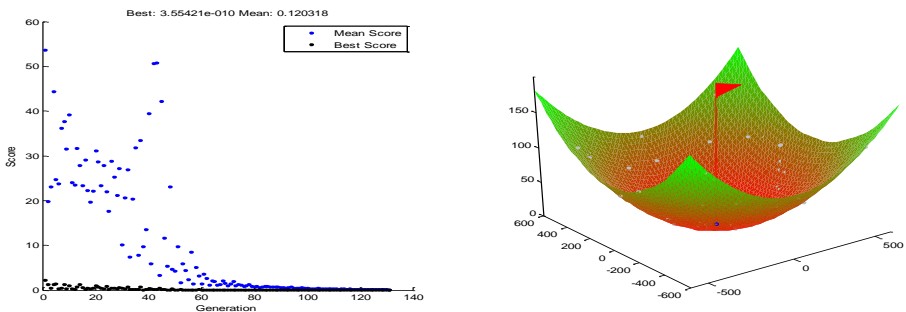

**Figure 27.** The experimental results of Griewangksfcn by C-MOPSO.

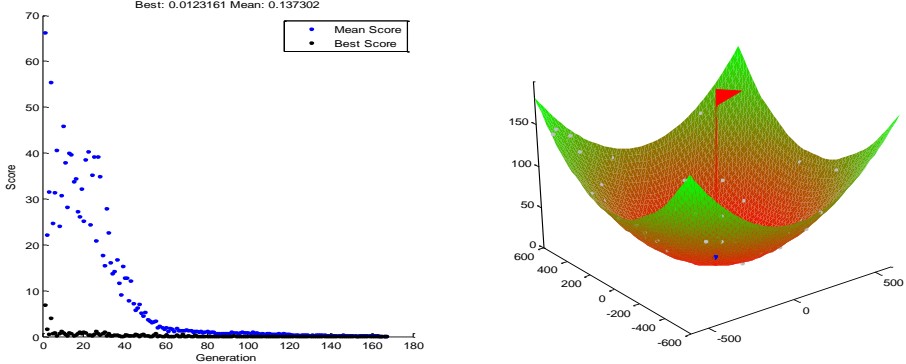

**Figure 28.** The experimental results of Griewangksfcn by E-MOPSO.

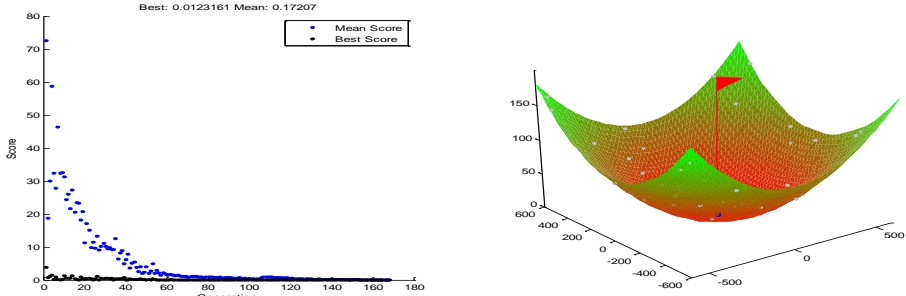

**Figure 29.** The experimental results of Griewangksfcn by MOPSO.

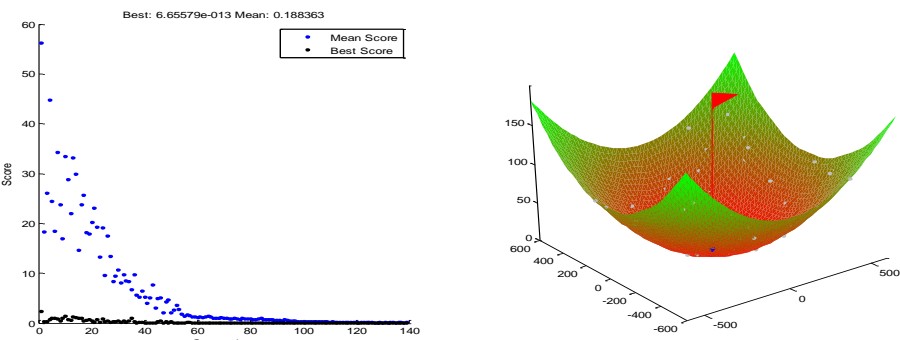

**Figure 30.** The experimental results of Griewangksfcn by PSO.

Based on the above classical multi-objective test function analysis, we used the same test function, such as Nonlinearconstrdemo, and the different hybrid particle swarm optimization algorithms. The experimental results of the same test function are described in Figures 31–35 as follows.

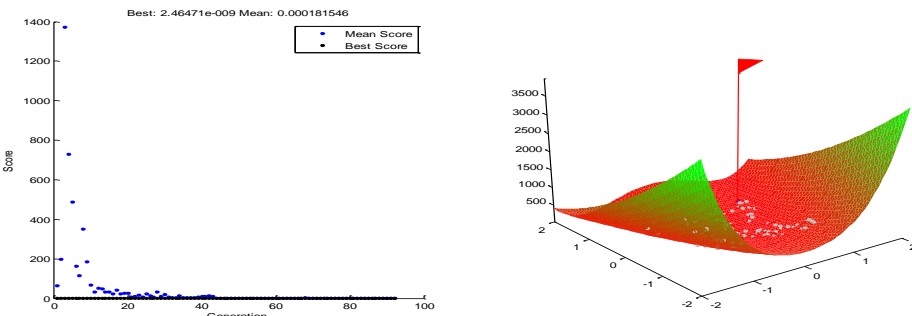

**Figure 31.** The experimental results of Nonlinearconstrdemo by HHCE-MOPSO.

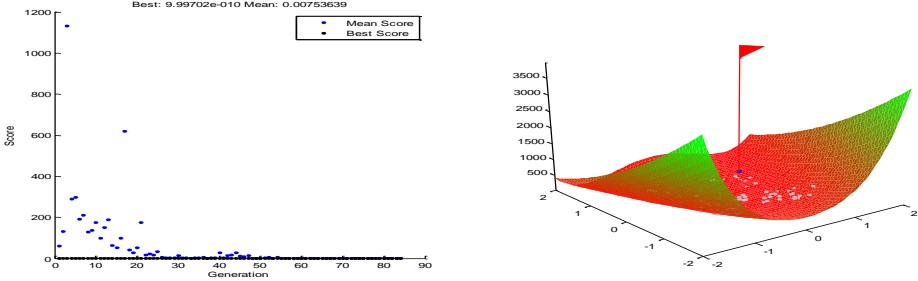

**Figure 32.** The experimental results of Nonlinearconstrdemo by C-MOPSO.

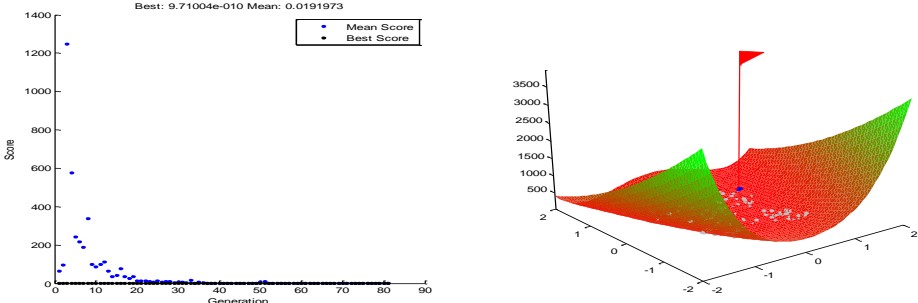

**Figure 33.** The experimental results of Nonlinearconstrdemo by E-MOPSO.

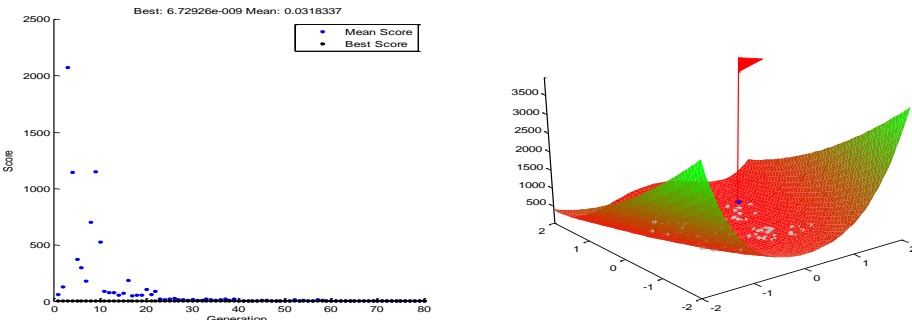

**Figure 34.** The experimental results of Nonlinearconstrdemo by MOPSO.

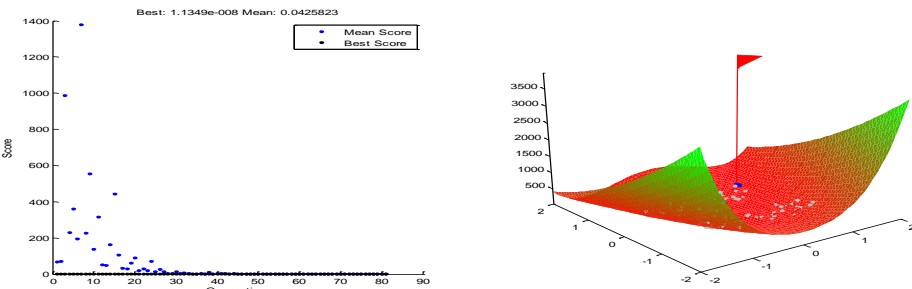

**Figure 35.** The experimental results of Nonlinearconstrdemo by PSO.

Based on the above classical multi-objective test function analysis, we used the same test function, such as Rastriginsfcn, and different hybrid PSO algorithms. The experimental results of the same test function are described in Figures 36–40 as follows.

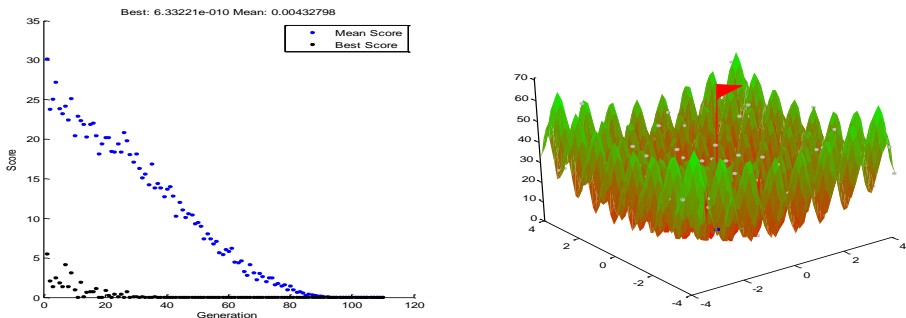

**Figure 36.** The experimental results of Rastriginsfcn by HHCE-MOPSO.

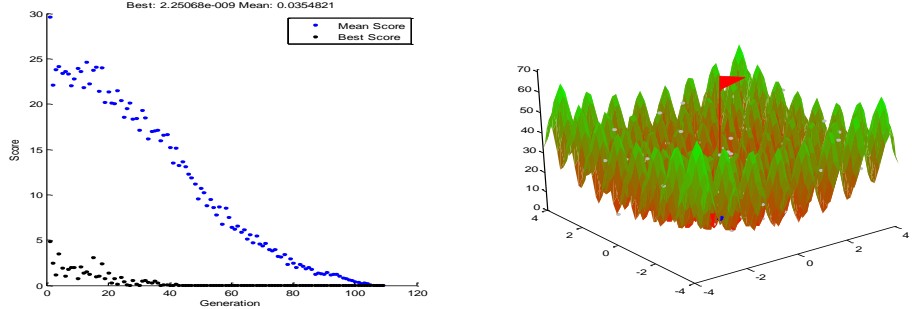

**Figure 37.** The experimental results of Rastriginsfcn by C-MOPSO.

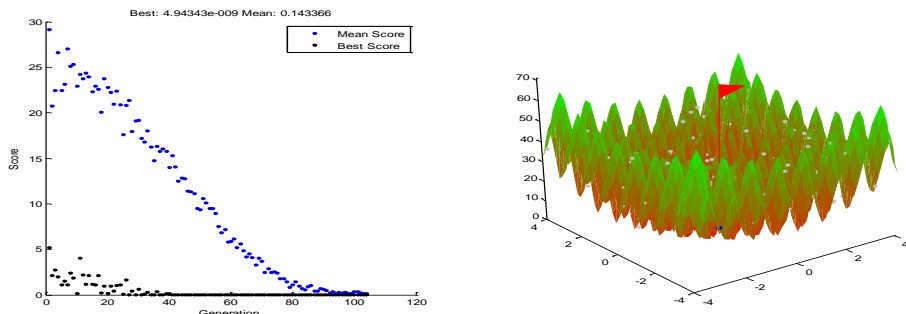

**Figure 38.** The experimental results of Rastriginsfcn by E-MOPSO.

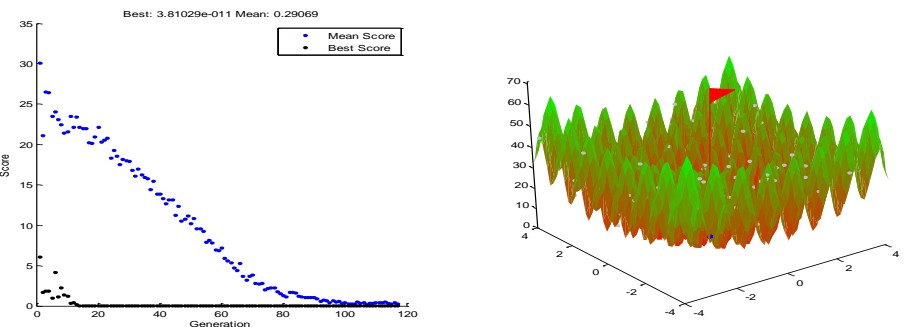

**Figure 39.** The experimental results of Rastriginsfcn by MOPSO.

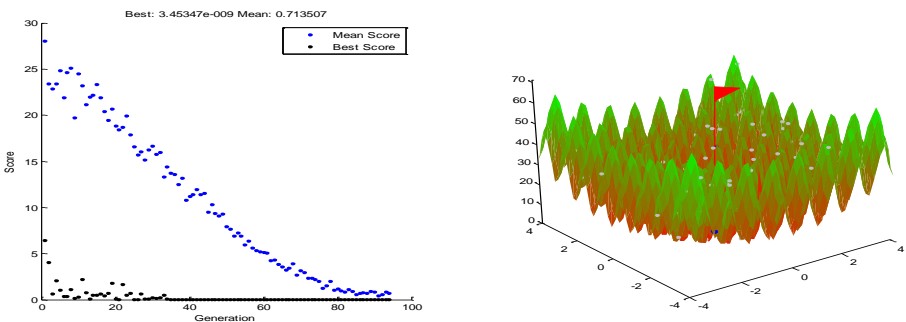

**Figure 40.** The experimental results of Rastriginsfcn by PSO.

Based on the above analysis, we will use the same test function, such as Schwefelsfcn, and different hybrid PSO algorithms. The experimental results of the same test function are described in Figures 41–45 as follows.

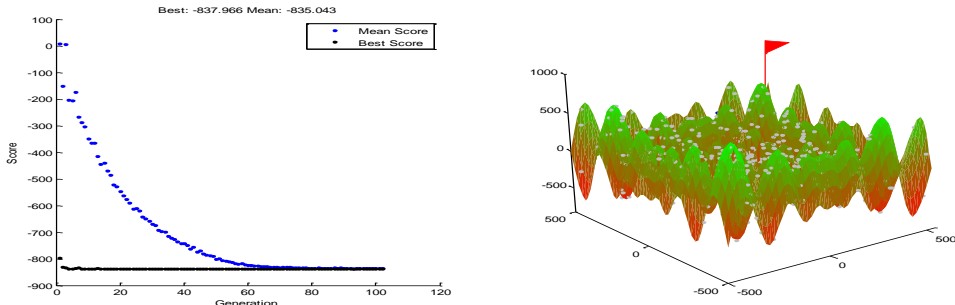

**Figure 41.** The experimental results of Schwefelsfcn by HHCE-MOPSO.

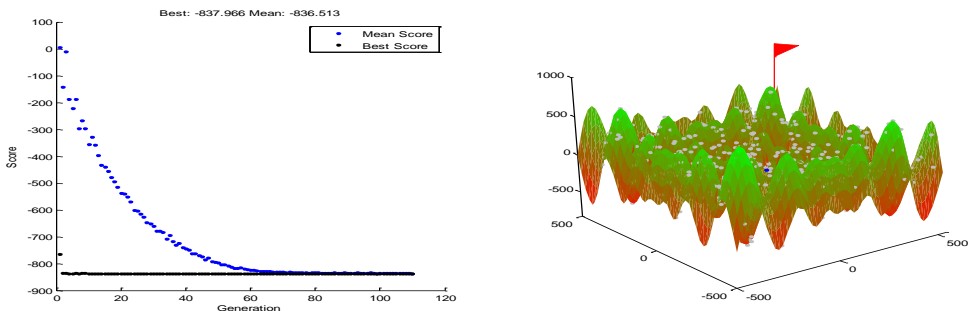

**Figure 42.** The experimental results of Schwefelsfcn by C-MOPSO.

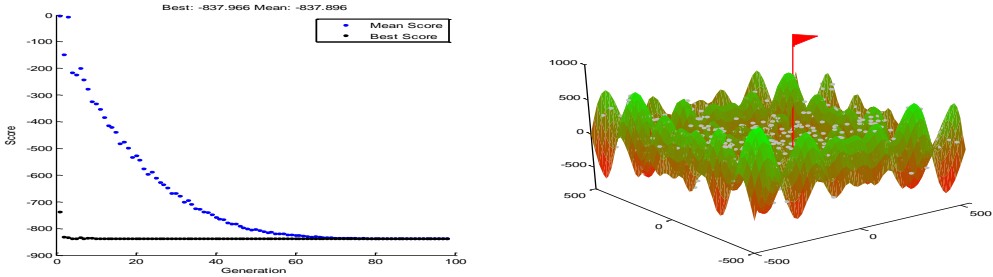

**Figure 43.** The experimental results of Schwefelsfcn by E-MOPSO.

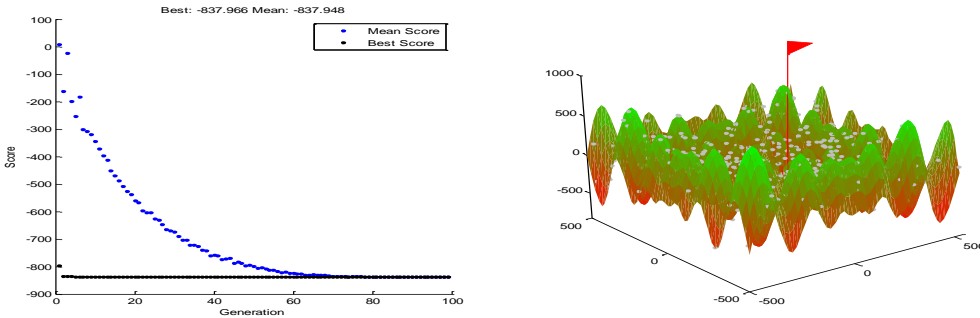

**Figure 44.** The experimental results of Schwefelsfcn by MOPSO.

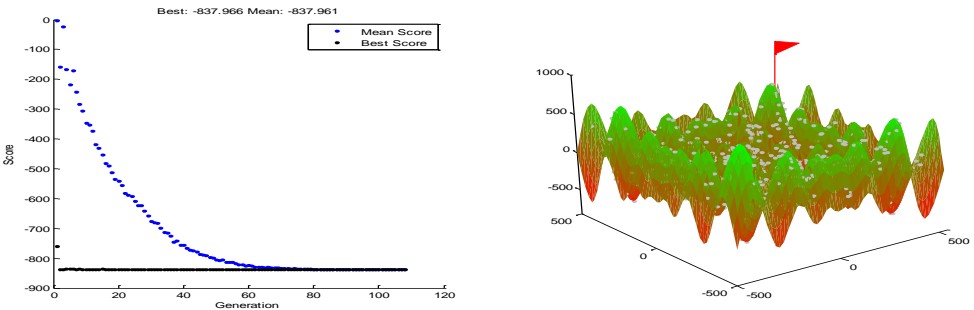

**Figure 45.** The experimental results of Schwefelsfcn by PSO.

From the experimental results of the classical multi-objective test function analysis, it can be seen that the algorithm is simple and effective, with a fast convergence speed and high accuracy. On the basis of theoretical analysis, we used the classic multi-objective test function to conduct effective experiments. Through experiments(all the results of test function were averaged on 50 independent runs), we found that the cloud model is an effective tool in uncertain transformation between qualitative concepts and the quantitative expressions. At the same time, the experimental results show that the algorithm can effectively solve the test problems of most continuous complex targets, which reflects the

feasibility of the hybrid algorithm. At the same time, the experimental results show that the algorithm still has a certain degree of robustness. Therefore, the multi-objective PSO algorithm can effectively solve complex multi-objective problems, and can be combined with various advanced constraint processing technologies to deal with single objective and multi-objective optimization problems.

To compare the performance results of hybrid algorithms, numerical and many segmentation measures have been discussed. The result of this implementation showed that the information entropy can be used to control the search, which will strengthen the evolutionary purpose contained in the particle swarm and make the particles converge to the optimal solution more quickly. Through a series of simulation experiments of the UCI dataset, the rationality and feasibility of the hybrid algorithm was also found to be statistically different to other algorithms are verified. Through experimental analysis, the performance parameter results of HHCE-MOPSO clustering algorithms are shown in Table 2 as follows.

**Table 2.** The performance parameter results of hybrid algorithms.

| Dataset | Algorithm | Accuracy | Sensitivity | Specificity | G-mean | AUC | MCC |
|---|---|---|---|---|---|---|---|
| Lenses | K-mean | 0.6200 | 0.5520 | 0.6215 | 0.4215 | 0.8525 | 0.8952 |
| | K-Boosted | 0.7520 | 0.6020 | 0.7010 | 0.6025 | 0.7980 | 0.8821 |
| | K-SMOTE | 0.8625 | 0.6250 | 0.8215 | 0.8520 | 0.8678 | 0.8150 |
| | SMOTE-Boosted | 0.8920 | 0.6525 | 0.8950 | 0.9010 | 0.8960 | 0.8320 |
| | HH-PSO-K | 0.9700 | 0.7020 | 0.9320 | 0.9250 | 0.9525 | 0.8980 |
| Lymphography | K-mean | 0.7150 | 0.6420 | 0.6315 | 0.5415 | 0.9025 | 0.9002 |
| | K-Boosted | 0.7620 | 0.6520 | 0.7216 | 0.6225 | 0.8180 | 0.8920 |
| | K-SMOTE | 0.8832 | 0.6750 | 0.8415 | 0.8820 | 0.8978 | 0.8255 |
| | SMOTE-Boosted | 0.8920 | 0.6820 | 0.9050 | 0.9100 | 0.9060 | 0.8610 |
| | HH-PSO-K | 0.9805 | 0.7120 | 0.9410 | 0.9360 | 0.9620 | 0.9080 |
| Arrhythmia | K-mean | 0.5600 | 0.5320 | 0.6015 | 0.5015 | 0.7525 | 0.8650 |
| | K-Boosted | 0.7120 | 0.6120 | 0.7210 | 0.5025 | 0.8080 | 0.8620 |
| | K-SMOTE | 0.8425 | 0.6040 | 0.8415 | 0.7520 | 0.8378 | 0.8050 |
| | SMOTE-Boosted | 0.8420 | 0.6425 | 0.8990 | 0.8080 | 0.8460 | 0.8220 |
| | HH-PSO-K | 0.9200 | 0.8020 | 0.9120 | 0.9150 | 0.9225 | 0.8900 |
| Bach-Chorales | K-mean | 0.8250 | 0.7520 | 0.8215 | 0.7200 | 0.8500 | 0.9052 |
| | K-Boosted | 0.8520 | 0.8080 | 0.8010 | 0.8024 | 0.8180 | 0.9021 |
| | K-SMOTE | 0.9025 | 0.8650 | 0.9215 | 0.8720 | 0.8878 | 0.8950 |
| | SMOTE-Boosted | 0.9120 | 0.8925 | 0.9250 | 0.9110 | 0.9060 | 0.8920 |
| | HH-PSO-K | 0.9800 | 0.9020 | 0.9420 | 0.9350 | 0.9625 | 0.9080 |

**Table 2.** *Cont.*

| Dataset | Algorithm | Accuracy | Sensitivity | Specificity | G-mean | AUC | MCC |
|---|---|---|---|---|---|---|---|
| Connect-4 | K-mean | 0.7800 | 0.5620 | 0.6415 | 0.5415 | 0.8825 | 0.9052 |
| | K-Boosted | 0.7920 | 0.6120 | 0.7110 | 0.6428 | 0.8280 | 0.8925 |
| | K-SMOTE | 0.8925 | 0.6350 | 0.8455 | 0.8625 | 0.8808 | 0.8250 |
| | SMOTE-Boosted | 0.9020 | 0.6690 | 0.9050 | 0.9110 | 0.9065 | 0.8420 |
| | HH-PSO-K | 0.9800 | 0.7125 | 0.9520 | 0.9350 | 0.9625 | 0.9080 |
| Covertype | K-mean | 0.6400 | 0.5880 | 0.6450 | 0.4452 | 0.8700 | 0.9085 |
| | K-Boosted | 0.7720 | 0.6250 | 0.7250 | 0.6228 | 0.8140 | 0.9020 |
| | K-SMOTE | 0.8825 | 0.6487 | 0.8430 | 0.8720 | 0.8245 | 0.8250 |
| | SMOTE-Boosted | 0.9150 | 0.6789 | 0.9120 | 0.9120 | 0.9005 | 0.8520 |
| | HH-PSO-K | 0.9800 | 0.7283 | 0.9410 | 0.9350 | 0.9500 | 0.9250 |
| Cylinder-Bands | K-mean | 0.6700 | 0.6030 | 0.6710 | 0.5780 | 0.9000 | 0.9250 |
| | K-Boosted | 0.8020 | 0.6528 | 0.7523 | 0.6521 | 0.8452 | 0.9030 |
| | K-SMOTE | 0.9010 | 0.6728 | 0.8780 | 0.9010 | 0.9060 | 0.8650 |
| | SMOTE-Boosted | 0.6250 | 0.7055 | 0.9200 | 0.9420 | 0.9250 | 0.8920 |
| | HH-PSO-K | 0.9360 | 0.7560 | 0.9500 | 0.9526 | 0.9602 | 0.9360 |
| Dermatology | K-mean | 0.4200 | 0.5320 | 0.6015 | 0.4012 | 0.8325 | 0.8700 |
| | K-Boosted | 0.5550 | 0.5820 | 0.6701 | 0.5890 | 0.7780 | 0.8620 |
| | K-SMOTE | 0.7625 | 0.6030 | 0.8016 | 0.8368 | 0.8470 | 0.8010 |
| | SMOTE-Boosted | 0.8020 | 0.6328 | 0.8750 | 0.8950 | 0.8762 | 0.8120 |
| | HH-PSO-K | 0.9020 | 0.7020 | 0.9020 | 0.9010 | 0.9325 | 0.8750 |
| Diabetes | K-mean | 0.6800 | 0.5960 | 0.6815 | 0.4815 | 0.9025 | 0.9450 |
| | K-Boosted | 0.8125 | 0.6720 | 0.7615 | 0.6620 | 0.9380 | 0.9320 |
| | K-SMOTE | 0.9225 | 0.6854 | 0.8915 | 0.9120 | 0.9278 | 0.8850 |
| | SMOTE-Boosted | 0.9320 | 0.7025 | 0.9450 | 0.9510 | 0.9365 | 0.8920 |
| | HH-PSO-K | 0.9800 | 0.7820 | 0.9520 | 0.9750 | 0.9825 | 0.9580 |
| Hayes-Roth | K-mean | 0.7000 | 0.6320 | 0.7015 | 0.6015 | 0.7825 | 0.9552 |
| | K-Boosted | 0.8325 | 0.6824 | 0.7800 | 0.6828 | 0.8585 | 0.9021 |
| | K-SMOTE | 0.9225 | 0.6858 | 0.9015 | 0.9120 | 0.9178 | 0.8950 |
| | SMOTE-Boosted | 0.9620 | 0.7325 | 0.9750 | 0.9710 | 0.9460 | 0.8910 |
| | HH-PSO-K | 0.9805 | 0.7820 | 0.9790 | 0.9750 | 0.9520 | 0.9785 |

Through experiments, we found that the performance of HHCE-MOPSO clustering algorithm could be evaluated using some synthetic and real-life datasets with some validity metrics (cluster numbers, intra cluster distance, intercluster distance, G-mean, and MCC), which was also compared with some prevalent state-of-art automatic clustering algorithms. Furthermore, we found that big data clustering is completed in a reasonable time and is sustainable as well over the dynamic data. The impact of well-defined distance functions on the objective function of different clustering algorithms has been compared with the results of the proposed algorithm. To further verify the hybrid algorithm, we used the UCI, such as Hayes-Roth, for experimental simulation and the results are shown in Figures 46–48 as follows.

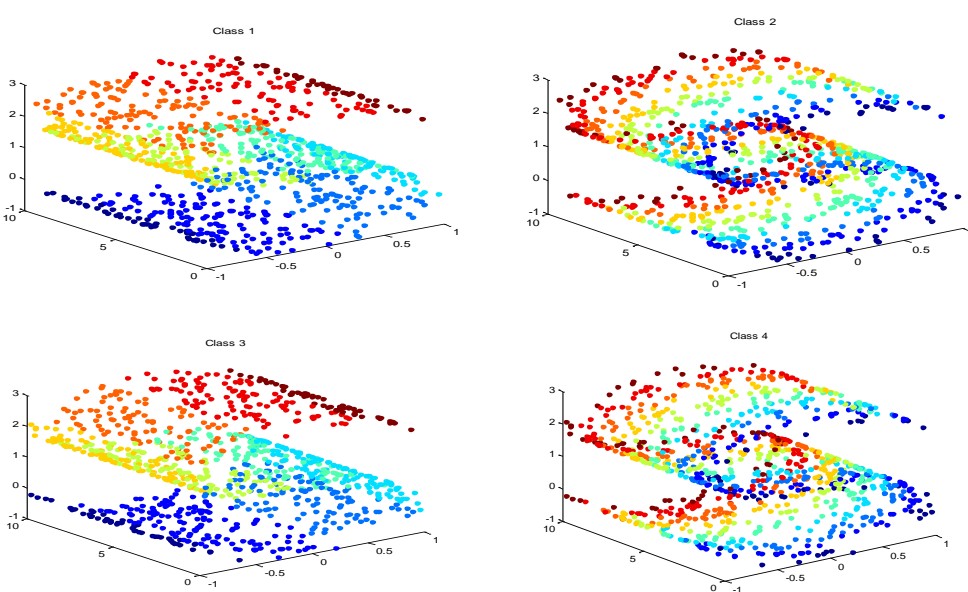

**Figure 46.** The simulationclustering results withPSO.

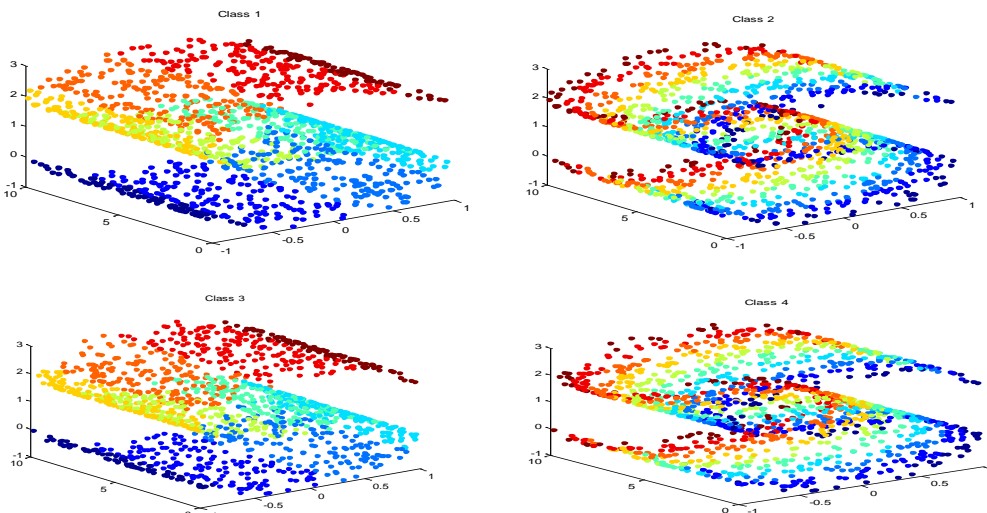

**Figure 47.** The simulationclustering results with MOPSO.

The experimental data analysis shows that the HHCE-MOPSO algorithm has good performance in terms of optimization quality and convergence speed. The more complex the attributes and higher the dimensions of the dataset, the worse the clustering effect will be if the same algorithm is applied. At the same time, the experiments on different scale datasets verify the effectiveness and efficiency of the hybrid clustering algorithm. Simultaneously, the hybrid PSO clustering algorithm proposed has good performance in different subsets of features in clustering accuracy and efficiency. Through the analysis of experimental data, the experimental results show that the proposed hybrid clustering algorithm is superior to the traditional clustering algorithm in clustering accuracy, and the running time under the same conditions is superior to the time of the traditional clustering algorithm. In addition, when the number of data samples increases, the advantage of the algorithm in time will gradually decrease. Although the hybrid clustering algorithm based on the entropy theory and cloud model is superior to other traditional clustering technologies, it still has some inevitable defects. Therefore, it is necessary to further study how to ensure the reasonable setting of relevant parameters in the future.

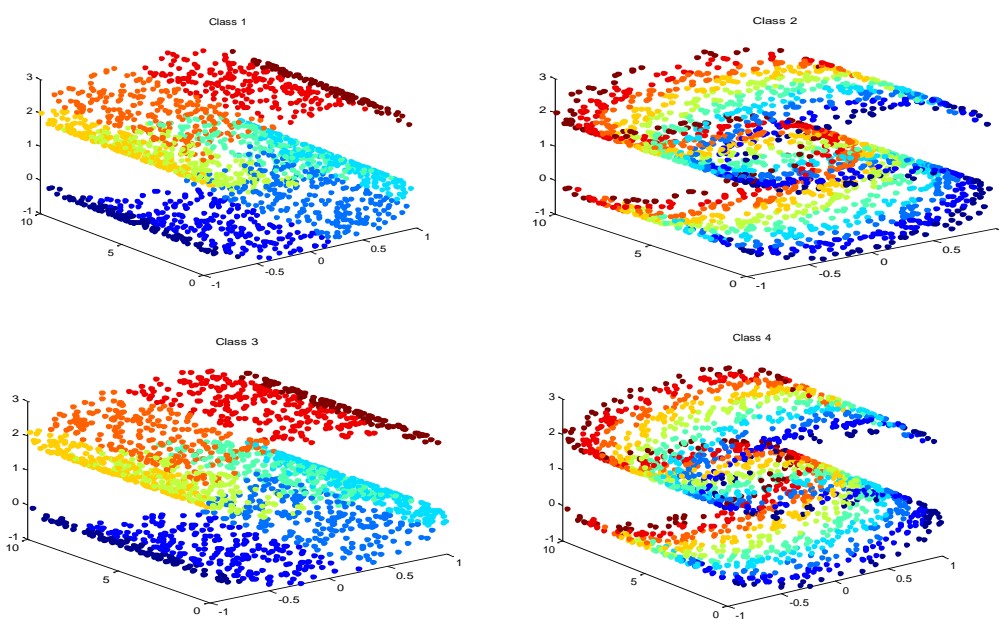

**Figure 48.** The simulationclustering results with HHCE-MOPSO.

## 5. Relevant Conclusions

A novel, hybrid, high-dimensional PSO clustering algorithm based on the cloud model and entropy theory is proposed by applying the cloud model and entropy theory to analyze the relationship between similarity measurement and data distribution. It can effectively eliminate the unimportant features in high-dimensional data sets, thus improving the performance of the hybrid clustering algorithm. The extensive experimental results on some classical benchmark functions demonstrate that the proposed hybrid algorithm produces encouraging results compared with the clustering techniques established in the literature. With the content, which is from general to complex and from unconstrained problems to constrained problems, we propose different HHCE-MOPSO algorithms that achieve a promising result by combining the excellent features, which was developed by hybridization of the PSO in this work. Through in-depth analysis, we found that the clustering algorithm is prone to premature or stagnation problems in the optimization process, which is mainly due to the insufficient speed updating ability of each particle in the late stage of PSO. In the meantime, we use the cloud model theory to realize the dynamic adjustment of the multi-rule uncertainty of the inertia weight, and the test results of test function show that this method has a fast convergence speed and a good optimization effect. Combined with frontier group intelligent optimization, the scope of multi-objective optimization problem is expanded. Subsequently, the obtained experimental results demonstrate that the hybrid algorithm is suitable for both scientific research and engineering applications, although there are also some shortcomings inevitably. Therefore, it is necessary to improve the proposed algorithm according to the characteristics of actual problems, so that it can better solve engineering problems with certain characteristics.

In summary, we have presented the HHCE-MOPSO algorithm. Through analysis, we found that it is a new idea to design a clustering algorithm to improve clustering quality and speed effectively. In addition, we also proved that the updating mechanism and the parameter design of the population mechanism of the multi-objective particle swarm optimization clustering algorithm are critical to obtain the optimal solution through a series of experiments. At the same time, we found that the similarity between objects is measured by information entropy and the similarity threshold is directly calculated by the data. In addition, information entropy clustering can be used to merge similar factors and can simplify the complexity of the system. Furthermore, a series of experiments showed that the HHCE-MOPSO algorithm, which takes advantages of the PSO, can achieve better clustering results and is feasible and effective. Hence, sustainability and dynamicity of

big data have been augmented by extending the HHCE-MOPSO algorithm to provide automatic clustering of the data. However, the data presents the characteristics of high dimensions and small samples with the advent of the era of big data. Therefore, we will need to further study a new feature selection clustering method suitable for high-dimensional and small sample imbalanced data. Although we have made some research achievements, unbalanced data clustering is a complex frontier problem. In the future, we should design an unbalanced clustering algorithm suitable for high-dimensional and small samples and improve the clustering effect and clustering performance by the spatial information.

**Author Contributions:** Conceptualization, R.-L.Z. and X.-H.L.; methodology, R.-L.Z.; software, R.-L.Z.; validation, R.-L.Z. and X.-H.L.; writing—original draft preparation, X.-H.L.; writing—review and editing, X.-H.L.; project administration, X.-H.L.; funding acquisition, R.-L.Z. and X.-H.L. All authors have read and agreed to the published version of the manuscript.

**Funding:** This research of work is supported by National Natural Science Foundation of China (Grant No. 72261005) and Guizhou science and technology planning projects (Grant No. ZK2021G339; ZK2022G080) and Guizhou philosophy and social science planning projects (Grant No. 21GZYB09; 21GZYB10) and the research base and critical special topics of think tanks (GDZX2021031, GDYB20210 22, GDYB2021023) and Guizhou Provincial Education Department Foundation (Grant No. 2022JD004) and the Talent Introduction Project of Guizhou University (Grant No. 2019016, 2019017, 20GLR001, 20GLR002) and 2022 National Social Science Fund Cultivation Project (GDPY2021014, GDPY2021015).

**Institutional Review Board Statement:** No applicable.

**Informed Consent Statement:** No applicable.

**Data Availability Statement:** No applicable.

**Conflicts of Interest:** The authors declare no conflict of interest.

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
