# Peer review of "A Novel Hybrid High-Dimensional PSO Clustering Algorithm Based on the Cloud Model and Entropy"

_applsci, doi:10.3390/app13031246_

Round 1
Reviewer 1 Report
The authors should provide the the feasibility of the hybrid PSO is verified by the simulation of the multi-objective test function and multi-dimensional cloud model and entropy theory.on related work and the current trend research. The authors should have enough background information to provide more acceptable of the proposed method in the paper.
Problem statement that the authors mentioned in the introduction section is quite weak and it cannot convince why we need the partitioned data sets. If the authors identify detailed background, it would make this paper more credible. If the author does so, it would streng then the position of the paper.
The authors need to illustrate that their proposed method can improve interactivity which the authors claimed that this issue is better than previous work that they reviewed. Rewrite the caption of figures. It is so small.
The author should mention how to solve the problem of the proposed method for the future research.
Reviewer 2 Report
The paper proposed a clustering algorithm for solving the problem of unbalanced data. However, the paper is not focusing on the unbalanced data. Most of the content are talking about the cloud model and PSO. Even the cloud model and PSO are not well depicted. The paper in its current form is not ready for further review and publication. And, the writing and organization of the paper also make it hard to recognize what the paper says.
Comments-1: there are quite a few places where grammatical and typographical errors need to be corrected before the manuscript is sent for publication
Comments-2: section 2 is not clear. Equations and symbols are not well explained, e.g., what is the meaning of x? At figure 2, what is the meaning of different colors? What is the axis x and y? At figure 1, what does the axis y mean?
Comments-3: pp. 5, line 138, what is the meaning of the sentence “However, due to the uncertainty and multidimensional attributes of multidimensional complex data.”?
Comments-4: pp. 5 lines 139 to 141, you have mentioned shortcomings of the clustering methods. Do you have any reference studies to support your opinion? For example
1. The sentence “The hybrid PSO algorithm has global optimization ability and distributed random search characteristics to solve the problem that traditional clustering algorithms are easy to fall into local optimization and sensitive to initial value.” Do you have any experiments or reference studies?
2. The sentence “Some scholars have proposed that the convergence speed and global optimization ability of the algorithm can be effectively improved through the combination of PSO which combines excellent features and traditional clustering methods”, which scholars or studies?
Comments-5: the paper gives many motivations and challenges. But, they are not reasonable. For example, from the sentence “The clustering algorithm has the ability of scalability, processing different types of data, finding clusters of arbitrary shape and processing high dimensional data”, how did you get such a challenge “Therefore, the emergence of large scale unbalanced data sets poses special challenges to clustering analysis technology”?
Comments-6: Do you have any real applications except simulations?
Comments-7: What is the test function Rosenbrock?
Reviewer 3 Report
Paper is well writen and discusses an important issue. Following are some minor corrections required before publication
1. write a bit enhanced abstract
2. if possible add clarity images
3. Add some latest references.
Round 2
Reviewer 2 Report
The comments are not well addressed. Please give the response to the comments one by one. Furthermore, the revised paper is still poorly written, especially lacking logicality.
Comments-1: The sentence “The hybrid PSO algorithm has global optimization ability and distributed random search characteristics to solve the problem that traditional clustering algorithms are easy to fall into local optimization and sensitive to initial value.” Do you have any experiments or referenced studies?
Please give the exact table or figure indices if you have experimental results. If there are referenced studies, please provide citations in your response.
Comments-2: The sentence “Some scholars have proposed that the convergence speed and global optimization ability of the algorithm can be effectively improved through the combination of PSO which combines excellent features and traditional clustering methods”, which scholars or studies?
The revised paper still starts with “some scholars”. Please give citations after “some scholars” as you mentioned in the response letter.
Comments-3: the paper gives many motivations and challenges. But they lack logicality due to poor writing. For example,
Comments-3.1: pp. 6, line 141, the sentence “However, due to the uncertainty and multidimensional attributes of multidimensional complex data” is not a complete sentence. You gave a “due to”, then what is the result?
Comments-3.2: in your revised sentence “At the same time, it can be found through experiments that this clustering should have computational efficiency and sustainability and be completed in the shortest time. Therefore, it is difficult to obtain high clustering accuracy by directly using existing machine learning algorithms and statistical analysis methods.” Is the use of the adverb “therefore” suitable here logically?
Author Response
Dear Reviewers:
Thank you for your letter and for the reviewers’ comments concerning our manuscript entitled “A Novel Hybrid High-dimensional PSO Clustering Algorithm Based on Cloud Model and Entropy”. Those comments are all valuable and very helpful for revising and improving our paper, as well as the important guiding significance to our researches. The authors would like to thank the referees and the Associate Editor for their patient comments to improve the quality of our manuscript. We have studied comments carefully and have made correction which we hope meet with approval. At the same time,we have revised the whole manuscript carefully and tried to avoid some grammatical, spelling or syntax errors. Revised portion are marked in red in the paper. In order to improve the quality of the paper, we feel that the reviewer's comments should be revised and improved again. At the same time, we are very grateful to the reviewers for their serious, responsible and professional attitude. Point by point responses to the reviewers’ comments are listed below this letter. Thank you very much for the comments and attitude of the reviewers, which have greatly improved the quality and level of our papers.Please see the attachment.
Once again, thank you very much for your comments and suggestions.
Looking forward to hearing from you.
Best regards.
Yours sincerely,
Ren-Long Zhang, Xiao-Hong Liu
